# Deep3DSIM: Super-resolution imaging of thick tissue using 3D structured illumination with adaptive optics

Jingyu Wang[1,2†], Danail Stoychev[1,2†‡], Mick A Phillips[1], David Miguel Susano Pinto[1,2], Richard M Parton[1], Nicholas James Hall[1], Joshua S Titlow[1], Ana Rita Faria[1], Matthew Wincott[2], Dalia Gala[1], Andreas Gerondopoulos[1], Niloufer Irani[1], Ian Dobbie[1*§], Lothar Schermelleh[1*], Martin J Booth[2*], Ilan Davis[1*‡]

[1]Department of Biochemistry, University of Oxford, Oxford, United Kingdom; [2]Department of Engineering, University of Oxford, Oxford, United Kingdom

*For correspondence:
ian.dobbie@jhu.edu (ID);
lothar.schermelleh@bioch.ox.ac.uk (LS);
martin.booth@eng.ox.ac.uk (MJB);
ilan.davis@glasgow.ac.uk (ID)

†These authors contributed equally to this work

Present address: ‡School of Molecular Biosciences, College of Medical, Veterinary & Life Sciences, University of Glasgow, University Avenue, Glasgow, United Kingdom; §Department of Biology, Johns Hopkins University, Baltimore, United States

Competing interest: The authors declare that no competing interests exist.

**Abstract** Three-dimensional structured illumination microscopy (3D-SIM) doubles the resolution of fluorescence imaging in all directions and enables optical sectioning with increased image contrast. However, 3D-SIM has not been widely applied to imaging deep in thick tissues due to its sensitivity to sample-induced aberrations, making the method difficult to apply beyond 10 μm in depth. Furthermore, 3D-SIM has not been available in an upright configuration, limiting its use for live imaging while manipulating the specimen, for example, with electrophysiology. Here, we have overcome these barriers by developing a novel upright 3D-SIM system (termed Deep3DSIM) that incorporates adaptive optics for aberration correction and remote focusing, reducing artefacts, improving contrast, restoring resolution, and eliminating the need to move the specimen or the objective lens in volume imaging. These advantages are equally applicable to inverted 3D-SIM systems. We demonstrate high-quality 3D-SIM imaging in various samples, including imaging more than 130 μm into the *Drosophila* brain.

## Editor's evaluation

Three-dimensional structured illumination microscopy (3D-SIM) is a technique that doubles imaging resolution. Still, its use has been limited by its sensitivity to aberrations in thick tissues and its lack of availability in an upright configuration. This convincing 'Deep3DSIM' method addresses these issues by using adaptive optics to correct aberrations and remote focusing for artefact-free volume imaging. This enables high-quality super-resolution imaging up to 130 μm deep into specimens, such as a *Drosophila* brain, while allowing for live-specimen manipulation, providing valuable advances on current efforts.

## Introduction

Fluorescence microscopy has greatly advanced with the development of super-resolution microscopy techniques that enable biological imaging with spatial resolutions well below the optical diffraction limit (*Huang et al., 2009*; *Sahl et al., 2017*; *Schermelleh et al., 2019*; *Schermelleh et al., 2010*). One of these approaches is 3D-SIM, which features twofold increased resolution in each spatial direction and enables fast imaging with optical sectioning for enhanced contrast. 3D-SIM recovers missing high spatial frequency information by illuminating the specimen with a fine sinusoidal pattern and computationally reconstructing an image with twice the resolution (*Gustafsson et al., 2008*; *Schermelleh*

*et al., 2008*). 3D-SIM has gained increasing popularity, not least due to its compatibility with standard fluorophores and labelling protocols commonly used in widefield or confocal microscopy.

Over the past decade, several off-the-shelf 3D-SIM instruments have become commercially available and successfully used for various biological applications. For example, the OMX, based on the original design by John Sedat (*Dobbie et al., 2011*), as well as a bespoke implementation for correlative 3D soft X-ray cryo-imaging on a synchrotron beamline (*Kounatidis et al., 2020*). However, they all share similar limitations that restrict their usage to thin specimens on inverted setups. Optical aberrations are one of the main contributors to reconstruction artefacts in 3D-SIM imaging (*Demmerle et al., 2017*). These include spherical aberrations induced by refractive index (RI) mismatches between the immersion media, mounting media, and sample, as well as sample-induced aberrations caused by RI inhomogeneities within the sample itself. The effect of these optical aberrations increases with imaging depth, making it difficult to obtain good 3D-SIM images at a depth greater than around 10 µm. It is also difficult to capture good images over larger z-ranges in most specimens when using oil-immersion objective lenses with high numerical aperture (NA) and limited working distance. Silicone oil and water immersion, or water-dipping, objective lenses, with lower NA, have longer working distances and reduced spherical aberration because of the better match to the RI of biological tissues. Currently, there are no upright off-the-shelf 3D-SIM systems that use water-dipping objective lenses, because such lenses are intended for imaging at depths far greater than 10 µm and because building 3D-SIM optics in an upright configuration presents complex engineering challenges. However, upright microscopy is essential for certain biological experiments, so 3D-SIM has not been applied to a variety of biological questions. For example, live imaging without a cover slip and with access to the sample for manipulations, such as microinjection and electrophysiology, has not been done with 3D-SIM.

One way to overcome the limitations described above is to incorporate adaptive optics (AO), using a deformable mirror (DM) in the optical path of a 3D-SIM system. AO works by changing the shape of the light's wavefronts, which are surfaces formed by points of equal phase, as the light wave propagates through space. AO can enable the correction of spherical and sample-induced aberrations and allow the rapid movement of the focus axially for volume imaging in a purely optical way (remote focusing), without the need to move the specimen or the objective lens. These functionalities have been described as prototype methods for widefield (*Kam et al., 2007*), 3D-SIM (*Lin et al., 2021*; *Thomas et al., 2015*; *Žurauskas et al., 2019*), and multiphoton imaging (*Žurauskas et al., 2017*). However, no off-the-shelf 3D-SIM instruments are designed to allow the inclusion of AO. Moreover, AO hardware and software controls are currently not available as simple modules that can be added to the optical path of commercial systems. While some exciting bespoke-built AO-SIM systems have previously been described (*Li et al., 2017*; *Lin et al., 2021*; *Thomas et al., 2015*; *Turcotte et al., 2019*), these proof-of-principle prototypes use AO for aberration correction only, and they are not suitable for applications requiring an upright optical configuration with multiple channels. To our knowledge, although DM-driven remote focusing has been used in simulation (*Kner et al., 2011*) and confocal (*Poland et al., 2008*) or multiphoton imaging (*Žurauskas et al., 2017*), it has not been implemented in any 3D-SIM systems. Image scanning methods have been developed to enhance the spatial resolution in imaging of thick samples using multi-point scanning (*York et al., 2013*; *York et al., 2012*), multiphoton excitation (*Ingaramo et al., 2014*), or AO (*Zheng et al., 2017*). These methods use physical or digital pinholes together with photon reassignment to improve the resolution by a factor of $\sqrt{2}$ or greater when applying deconvolution. 3D-SIM achieves higher resolution, but manages out-of-focus background less efficiently than image scanning methods. The main reasons for this are the retention of the shot noise of the background light after reconstruction, and the reduced modulation contrast of the structured illumination (SI), potentially leading to reconstruction artefacts. These problems are compounded by depth and fluorescence density.

Here, we describe our bespoke upright Deep3DSIM prototype system with integrated AO, enabling deep imaging in live wholemount biological specimens with direct access for their manipulation (*Figure 1*). The system is based around a 60×/1.1 NA water-immersion objective lens, with a correction collar that allows its use in a water-dipping configuration without a cover slip. We demonstrate high-quality 3D-SIM with nearly twofold spatial resolution extension in three dimensions at wide depths from a few micrometres to 130 µm. We use AO not only for sample-induced aberration correction in deep tissue imaging but also for remote focusing. The latter enables fast transitions of the imaging plane and prevents pressure waves caused by the motion, along the optical axis, of the

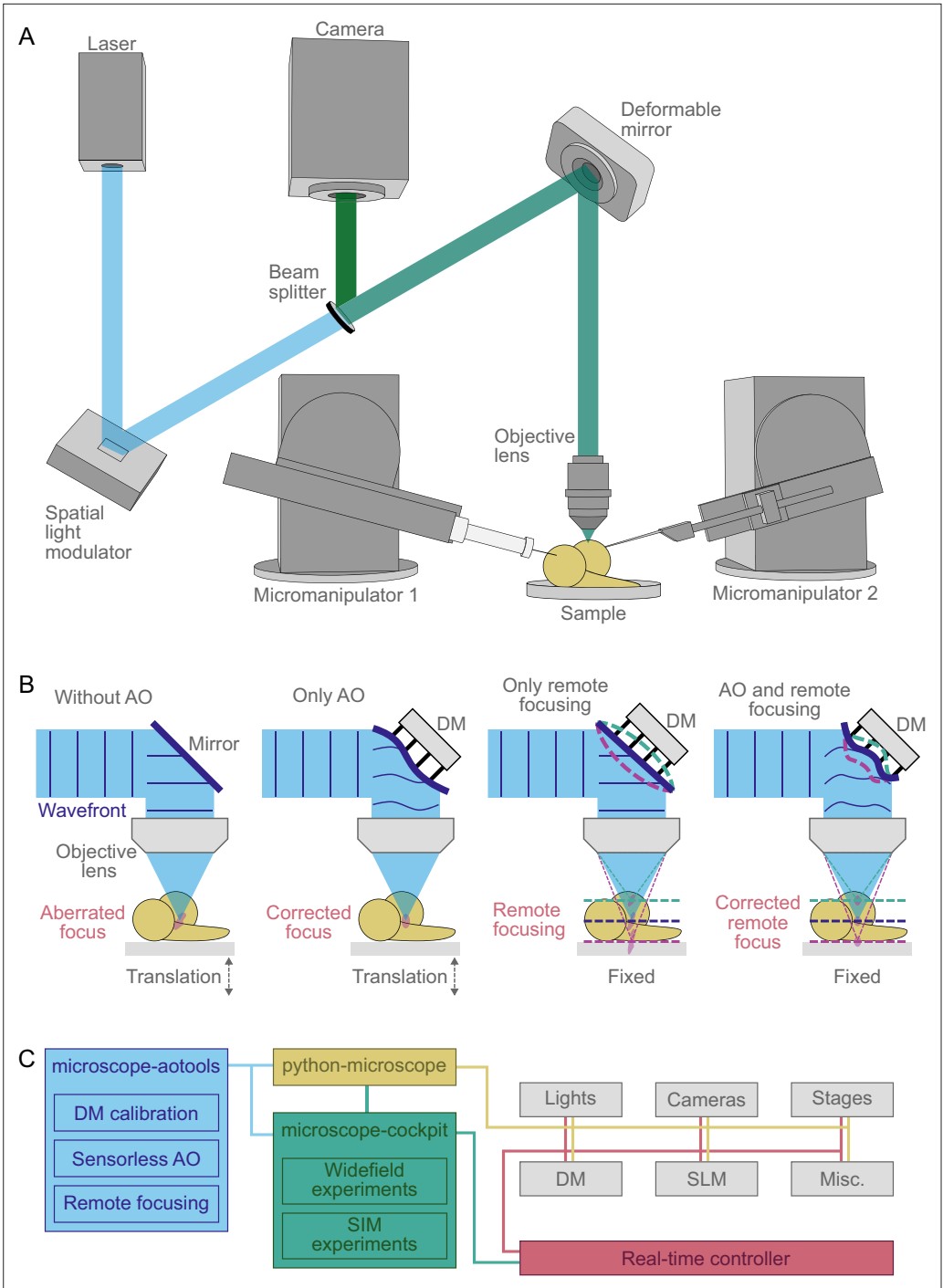

**Figure 1.** Simplified overview of the Deep3DSIM setup. (**A**) Optical arrangement of Deep3DSIM and the conceptual use of micromanipulators for applications such as microinjection and electrophysiology. The excitation light hits the spatial light modulator (SLM), which is conjugated to the object/image plane, and then reflects off a deformable mirror (DM) before being focused on the sample with an objective lens. In the imaging path, the reflected fluorescence is coupled off via a dichroic beam splitter and then collected by separate cameras for each channel. (**B**) Different imaging modes are enabled in parallel on Deep3DSIM. Deep imaging without adaptive optics (AO) usually leads to an aberrated point spread function (PSF) due to refractive index (RI) mismatch and sample inhomogeneity. Sensorless AO correction compensates for sample-induced aberrations. Remote focusing enables fast focusing at various depths without moving the specimen/objective. Combination of AO and remote focusing produces aberration-corrected imaging at different depths, without mechanical movement. (**C**) Deep3DSIM is controlled by Cockpit, which consists of three packages: python-microscope is responsible for

*Figure 1 continued on next page*

*Figure 1 continued*

the control of all hardware devices; microscope-cockpit provides a user-friendly GUI; microscope-aotools provides the AO functionality. The system uses a real-time controller, in this case, the Red Pitaya STEMlab 125–14, to coordinate different instruments for image acquisition with structured or widefield illumination.

specimen or the objective lens. Our novel approach and instrument design enable simultaneous multi-channel imaging in conventional and super-resolution modes. The control of Deep3DSIM is based on our previously published user-friendly open-source Python software named Cockpit (see Methods and materials), controlling the high-performance hardware devices required for 3D-SIM, while achieving highly accurate and precise timing at fast rates. We first test the system's performance using beads and cells in culture. We then apply the full range of novel imaging modalities available on Deep3DSIM to a wide range of specimen types, from mammalian tissue culture cells to *Drosophila* larval brains and embryos. Finally, although our use of AO for aberration correction and remote focusing is demonstrated on an upright microscope, the principles we have established could be applied equally to an inverted 3D-SIM system. Use of AO on an inverted system, like on an upright one, would provide considerably extended depth of imaging and rapid acquisition of 3D volumes without moving the specimen or the objective lens.

## Results

To initially assess the baseline performance of our Deep3DSIM system, we imaged green fluorescent latex microspheres ('beads') with an average diameter of 100 nm, attached to a glass cover slip. We used the objective lens in a water-immersion configuration, imaging through the cover slip, to make the assessment comparable to common experimental routines. We found the resolution of Deep3DSIM to be a little lower than double that of widefield. 3D images of the beads were used to estimate the point spread function (PSF) of the system in widefield (WF) and 3D-SIM modalities. The PSF provides a complete characterisation of the optical system. We fitted the intensities of these images to Gaussian curves, both laterally and axially, and then used the full width at half-maximum (FWHM) of the Gaussian curves to estimate the lateral and the axial resolution. We repeated these calculations for multiple beads for each modality (*Figure 2A*). The mean lateral resolution was 185 nm (standard deviation (SD) of 13 nm) for 3D-SIM and 333 nm (SD = 9 nm) for widefield, whereas the mean axial resolution was 547 nm (SD = 34 nm) and 893 nm (SD = 41 nm), respectively. We found that Deep3DSIM produced a resolution improvement that approaches but does not quite reach the theoretically possible doubling of the widefield resolution. This limitation is partly due to our choice of conservative line widths for the SI patterns (see Appendix 1 for details).

3D-SIM, like other super-resolution methods, is sensitive to even low levels of system and sample-induced aberrations, which are typically encountered in thin samples, such as single cells in culture. Such minor aberrations are often overlooked in conventional microscopy modalities such as widefield and confocal (*Wang and Zhang, 2021*). Therefore, in addition to the FWHM measurements of beads, we also estimated the resolution enhancement by 3D-SIM in mammalian tissue culture COS-7 cells in two channels. We used immunofluorescence to label microtubules with an antibody conjugated to Alexa Fluor (AF) 488 (*Figure 2B*, left) and, in another example, endoplasmic reticulum labelled with antibodies against Rtn4 and conjugated to AF 555 (*Figure 2B*, right). Both structures consist of tubules that are below the diffraction limit and the resolving power of the objective lens, which makes them highly suitable for testing the performance of super-resolution microscopy. We compared the 3D-SIM images to their pseudo-widefield (PWF) equivalents, which were obtained using SIMcheck (*Ball et al., 2015*) by averaging the raw SI images. Compared to the corresponding PWF images, the optical sectioning capability of 3D-SIM was easily noticeable by the large difference in out-of-focus signal, especially in the cross-section views. We estimated the resolution with analysis in frequency space by finding the support of the optical transfer function (OTF). The OTF is the Fourier transform of the PSF, and, just like it, it provides a model of the optical system. We used the contours of the OTF support (dashed lines in *Figure 2B*) as estimations of the resolution. In the green channel, the resolutions were 356 nm and 190 nm laterally, and 1009 nm and 568 nm axially, respectively for PWF and 3D-SIM. These resolution values were close to, and agreed with, the FWHM values measured from beads (*Figure 2A*). The performance of the system in the red channel was similar, but the spatial

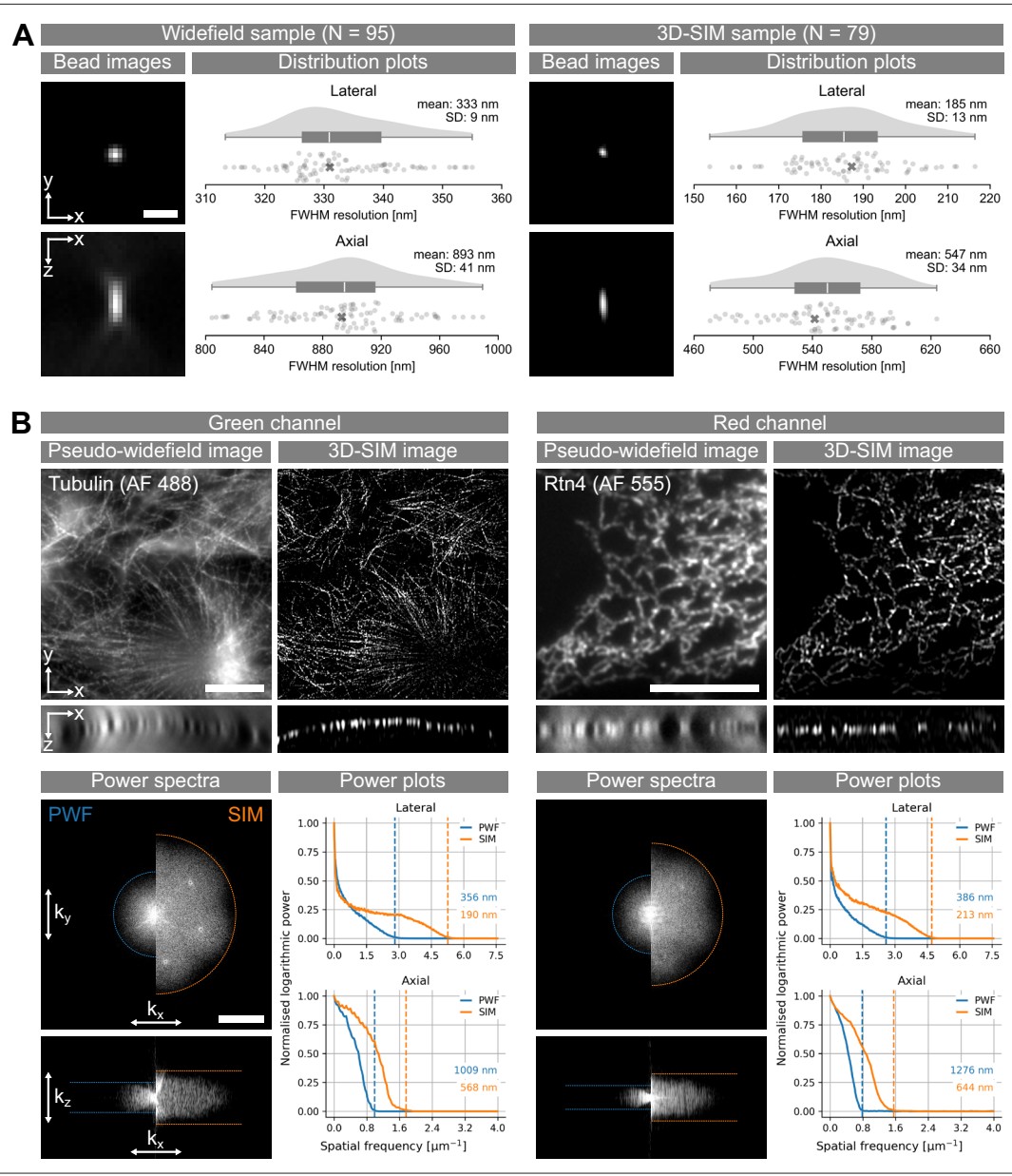

**Figure 2.** Resolution estimations of the Deep3DSIM system. (**A**) Full width at half-maximum (FWHM) measurements of 100 nm green fluorescent beads, deposited on a glass cover slip and imaged in water-immersion configuration. The images show lateral (XY) and axial (XZ) views of one bead from each sample, indicated with a cross on the distribution plots. SD: standard deviation. Scale bar: 1 µm. (**B**) Frequency space analysis of images of fixed COS-7 cells, with labelled microtubules in the green channel and endoplasmic reticulum in the red channel. The images were acquired in water-immersion configuration, imaging through a cover slip. Scale bars: 10 µm. The XY images are average projections along Z, and the XZ images are orthogonal projections along the middle of the Y axis. The power spectra are in logarithmic scale and centred on the zero frequency; they show lateral $k_{xy}$ view (top panel) and an axial $k_{xz}$ view (bottom panel) of the 3D DFT, both taken along the middle of the corresponding third axis. Scale bar: 3 µm$^{-1}$. The power spectra were thresholded to remove noise. The power plots show the attenuation of the frequency response with increasing resolution. The lateral power plot was created by radial averaging of the frequency amplitudes in the $k_{xy}$ view, while the axial plot shows the frequency amplitudes of the $k_{xz}$ view after averaging the two directions $0 \rightarrow \pm k_z$ and then averaging along the $k_x$ axis. The resolution thresholds (dashed lines) were chosen as the points at which the normalised logarithmic power reached 0.01 (i.e. 1%). The resolution thresholds were converted to the spatial domain by taking the inverse of the spatial frequency.

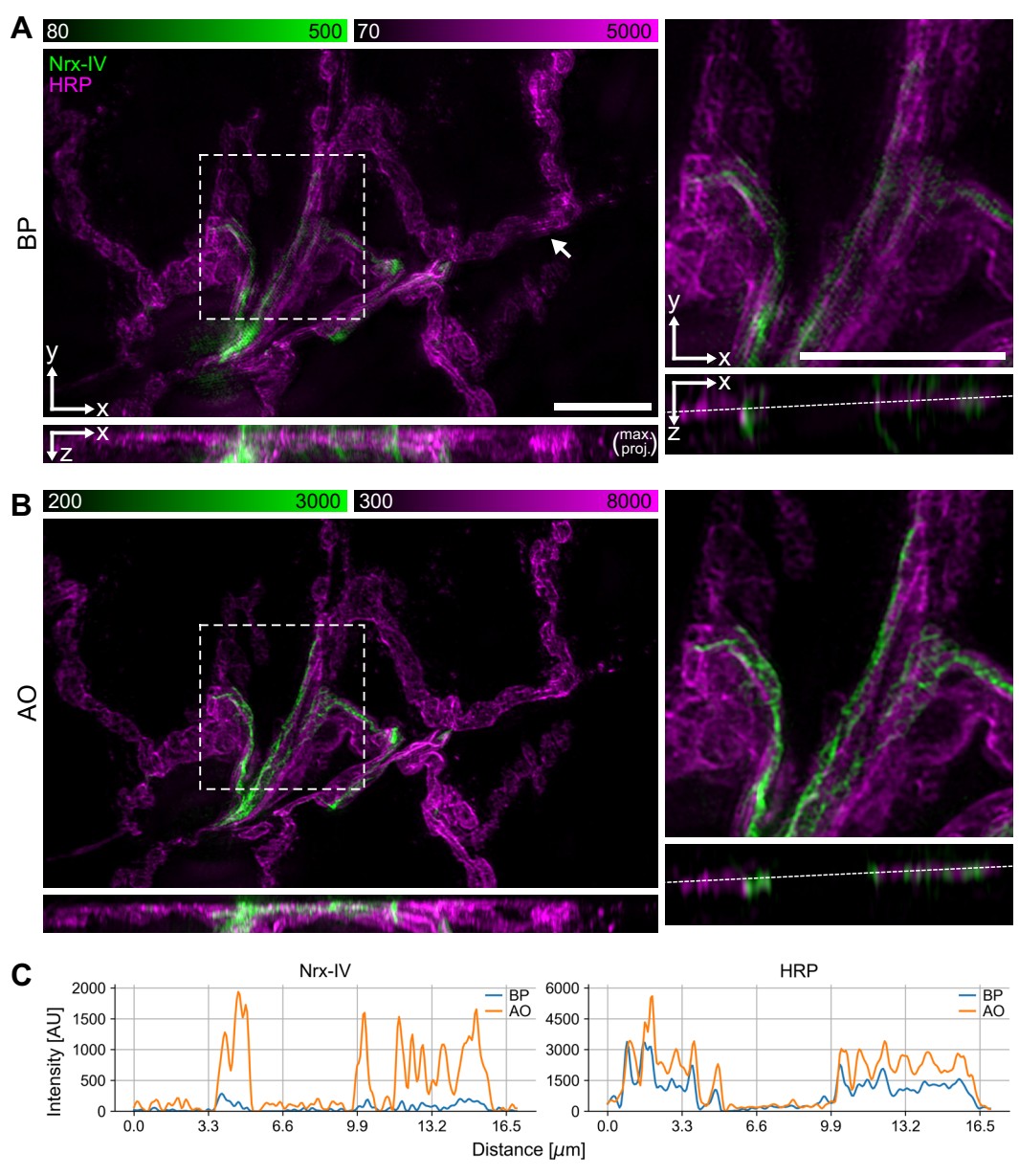

**Figure 3.** Adaptive optics (AO) correction removes artefacts and improves the contrast and resolution of 3D structured illumination microscopy (3D-SIM) in tissue imaging. (**A, B**) 3D-SIM images of neuromuscular junction in a fixed L3 *Drosophila* larva expressing Nrx-IV::GFP (green) and membrane labelled with anti-HRP antibodies (magenta) acquired without (**A**) and with AO (**B**), respectively. The data was acquired in water-immersion mode, imaging through a cover slip. The XZ views on the left are maximum projections along the Y axis. The boxed regions are shown scaled up on the right, with their own orthogonal XZ views along the middle of the Y axis. The arrow in (**A**) shows an example of ghosting artefacts in the red channel. The scale bars are 10 μm. (**C**) Intensity profiles along the dashed lines in the XZ views on the right of (**A**) and (**B**).

The online version of this article includes the following figure supplement(s) for figure 3:

**Figure supplement 1.** Frequency-space analysis and aberration correction.

frequency response was attenuated because of the longer wavelength. Here, the resolution values estimated from the power plots in PWF and SIM modes were 386 nm and 213 nm laterally, and 1276 nm and 644 nm axially, respectively. We performed a similar analysis for most of the subsequent experiments (supplements to *Figure 3*, *Figure 4* and *Figure 5*).

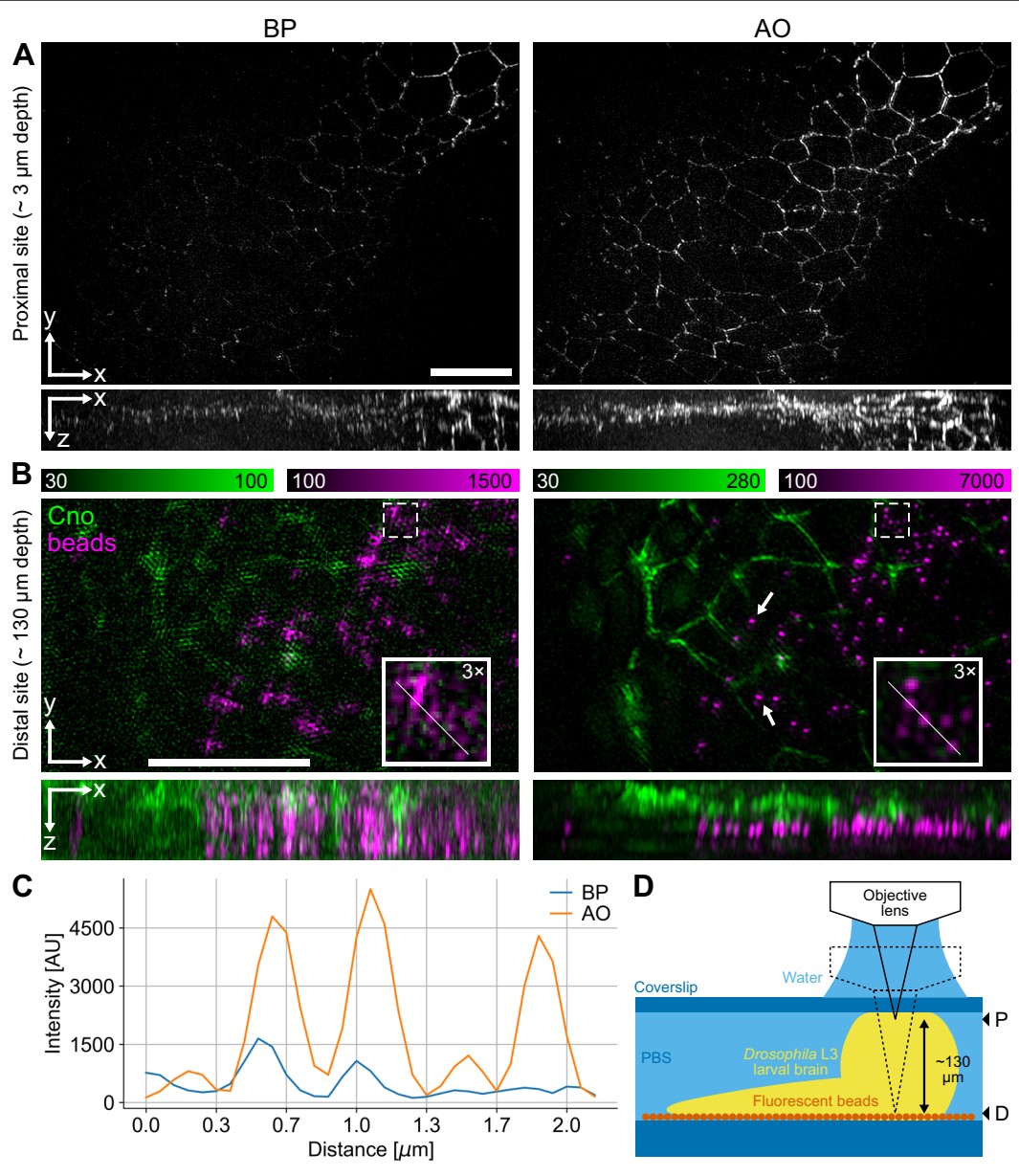

**Figure 4.** 3D structured illumination microscopy (3D-SIM) in deep tissue samples enabled by adaptive optics (AO) aberration correction. (**A**) 3D-SIM image stack of fixed *Drosophila* L3 larval brain expressing Cno::YFP and imaged from top downwards at a depth of ~3 µm from the cover slip surface without AO (left) and with AO (right). Both the lateral views (XY) and orthogonal views (XZ) show maximum projections. Scale bar: 10 µm. (**B**) 3D-SIM images of the same fixed *Drosophila* L3 larval brain, acquired through the entire volume of a single brain lobe at ~130 µm from the top surface, with 100 nm diameter red fluorescent beads attached to the surface of the glass slide. Images are displayed as in (**A**) but magnified. The insets show further 3× magnification of the regions indicated with dashed lines. Scale bar: 10 µm. (**C**) Intensity profiles of the red channel along the lines in the insets. (**D**) Schematic illustration of the specimen mounting and imaging. The larval brain was mounted in PBS between a glass cover slip (top) and a glass slide (bottom); the slide was coated with red fluorescent beads at medium density. Image stacks in (**A**) and (**B**) were recorded at the proximal (P) and distal (D) sites indicated with arrowheads on the right, respectively.

The online version of this article includes the following figure supplement(s) for figure 4:

**Figure supplement 1.** Frequency-space analysis and aberration correction.

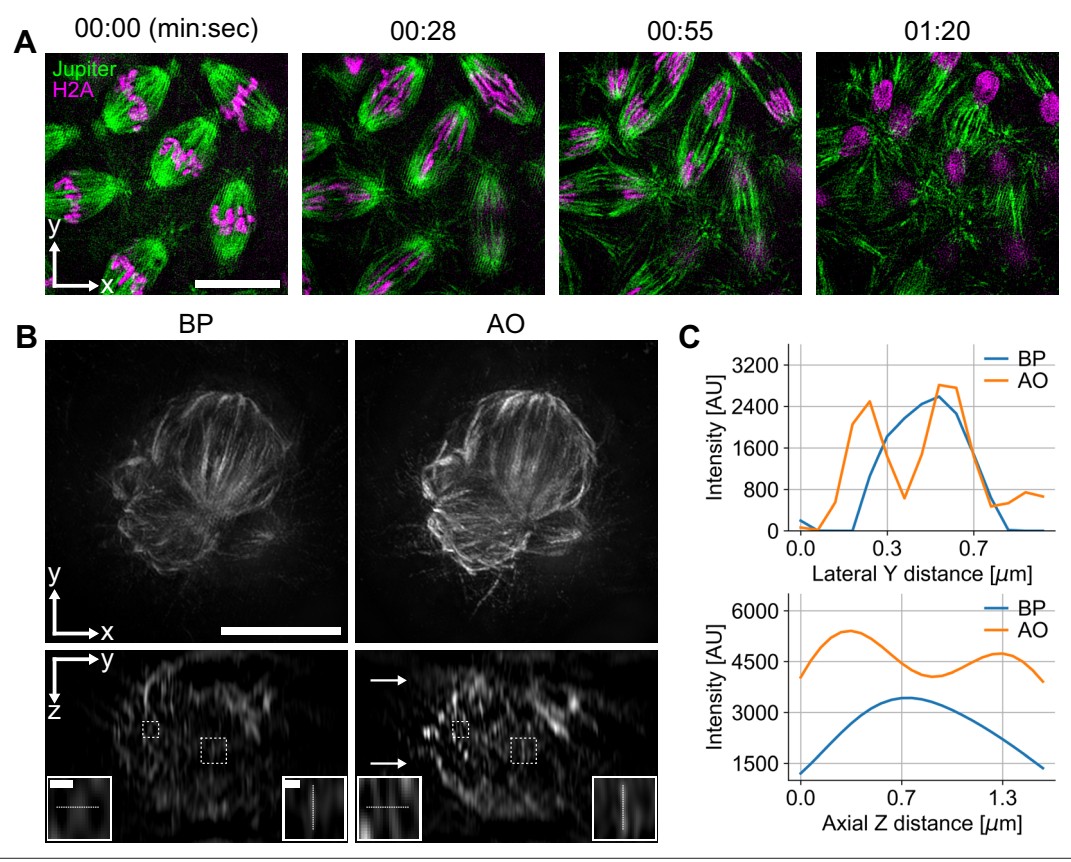

**Figure 5.** Remote focusing for fast live 3D structured illumination microscopy (3D-SIM) time-lapse imaging and for large volume imaging using multi-position aberration correction (MPAC). (**A**) Mitotic embryonic divisions in *Drosophila* syncytial blastoderm embryos, expressing transgenic Jupiter::GFP labelling microtubules (green) and transgenic Histone H2A::RFP labelling chromosomes (magenta). The imaging was carried out in water-dipping configuration, without the use of a cover slip. Images show maximum projection of volumes with about 1 μm thickness. Brightness and contrast of each image were adjusted independently. Scale bar: 10 μm. (**B**) COS-7 cell in metaphase with microtubules immunostained with AF 488, visualising the mitotic spindle. The volumes were acquired with remote focusing, imaging through a cover slip in water-immersion configuration. The lateral (XY) views are maximum projections. Aberration correction was performed at two imaging planes, indicated by arrows in the adaptive optics (AO) YZ view; dynamic correction was applied for all other planes. Insets show magnifications of the regions indicated with dashed boxes. Scale bars: 10 μm in the lateral bypass (BP) view and 0.5 μm in the insets. (**C**) Intensity plots in the lateral and axial directions, along the dashed lines in the insets, showing increased resolution in the AO (MPAC) case.

The online version of this article includes the following figure supplement(s) for figure 5:

**Figure supplement 1.** Multi-position aberration correction (MPAC) mode amplitudes.

Deep3DSIM is an upright system with a long working distance objective lens, in contrast to commercial 3D-SIM systems, which are built around inverted microscopes, and which are usually restricted to imaging thin samples, such as cultured cells. Deep3DSIM is particularly suitable for tissue-based imaging while manipulating the specimen with methods such as electrophysiology. We evaluated these unique features of the instrument by imaging the *Drosophila* neuromuscular junction (NMJ), a sample well suited for electrophysiological experiments. For convenience, we prepared fixed NMJ preparations by dissecting open the larvae, removing the guts, and pinning out the muscle layer with its associated motor neuron system to produce a so-called NMJ fillet preparation. These kinds of samples typically exhibit a substantial amount of RI inhomogeneity and mismatch, and they often require imaging deep into the specimen to examine specific sites of muscle innervation. These conditions led to a significant amount of optical aberrations, further emphasised by our use of a glycerol-based mounting medium (Vectashield). We used 3D-SIM with AO to image the dissected larva fillet

preparation from a transgenic line expressing a bright GFP protein trap of the cell-cell 'septate' junction protein Neurexin IV (Nrx-IV), at a depth of 5–15 µm (*Figure 3*). Our implementation of the AO did not use a dedicated wavefront sensor, instead relying on an indirect sensorless approach. We used a bypass (BP) mode as control, to compare between imaging with or without AO, where we switched the DM for a flat mirror. The images of the AO-corrected volume showed significant improvements in both intensity and contrast, compared to the images of the uncorrected volume (*Figure 3C*). The improvement was especially noticeable between the XZ views, as the major effect of aberrations is to distort the PSF along the optical axis. This was most notable for spherical aberrations, which elongate the focus along the optical axis.

To further test the microscope's performance at greater depth, we imaged, dissected, and subsequently formaldehyde-fixed whole *Drosophila* larval brains. We imaged both the top and bottom layers of the thickest part of the brain lobes, mounted between a standard microscope cover slip (#1.5) and a slide, providing a total imaging depth of ~130 µm (*Figure 4C*). To help visualise the effects of aberrations and to evaluate the ability of AO to correct them at depth, we placed 100 nm diameter red fluorescent beads (580/605 nm) on the glass slide underneath the tissue (*Figure 4D*). We acquired two image stacks centred on two sites – one proximal site ~3 µm away from the cover slip and another distal site ~130 µm away from it, which required focusing through the entire volume of the mounted brain (*Figure 4D*). The larvae used to prepare the samples came from a fly line with a YFP protein trap for *canoe* (*cno*), which encodes a protein that is expressed in the adherens junctions of the monolayer of neuroepithelial stem cells in the outer proliferation centres in the optic lobes. The protein expression marked the outlines of the cells and showed the characteristic geometrical mosaic pattern that emerges from the packing of epithelial cells, covering large parts of each brain lobe and thus providing a well-defined structure that could be imaged at various depths. We found that AO aberration correction made a small but distinct improvement to the contrast in the images of the proximal side of the brain lobe (*Figure 4A*). On the distal side, control images bypassing the AO suffered from severe aberration-induced reconstruction artefacts, with the cell outlines not observed as continuous structures (*Figure 4B*, left). In contrast, imaging at the same depth with AO correction enabled 3D-SIM reconstructions to produce clearer 3D images with reduced artefacts and enhanced contrast. Large areas of the cell outlines could be visualised (*Figure 4B*, right). The effect of aberrations and their correction is most evident in the red channel, where the images of the individual beads closely mimic the shape of the system's PSF, while the equivalent images without AO result in large unrecognizable fuzzy patches, instead of sharp spots. While the AO corrected images from the distal site showed minor residual aberrations (see Discussion for details) and reconstruction artefacts (see Methods and materials for further information), the overall PSF shapes were still dramatically improved with AO compared to the equivalent without AO. Moreover, the peak intensity and signal-to-noise ratio (SNR) with AO are also significantly higher than the results without AO, and the lateral FWHM resolution is preserved even at this depth (*Figure 4C*).

A key feature of Deep3DSIM is the ability to perform remote focusing. On the other hand, the reconstruction process of 3D-SIM is very stringent about the image quality and the image acquisition parameters, such as the accuracy and precision of Z steps in the acquisition of Z-stacks. Therefore, we wanted to show that remote focusing can meet the strict requirements of 3D-SIM reconstruction, and that the remote focusing can be used as a direct replacement of a mechanical Z stage over a moderate range. We demonstrate this with two examples, starting first with live imaging. Living specimens already present several challenges for 3D-SIM imaging on their own (*Gao et al., 2012*). Particularly important are the acquisition speed, which can lead to motion artefacts if not sufficiently high, and photobleaching and phototoxicity, which occur when the rate of exposure and the intensity of the excitation light are too high. The Deep3DSIM approach has the potential to resolve both these challenges by using remote focusing for fast image acquisition while preserving the high image quality of the system. To show this, we imaged *Drosophila* embryos undergoing rapid mitosis, where each division cycle lasts just a few minutes (*Figure 5A*). We collected eggs from transgenic animals expressing the microtubule-binding protein Jupiter fused to GFP and the histone H2A proteins fused to RFP. The Jupiter::GFP H2A::RFP eggs were dechorionated and mounted in aqueous solution in order to image chromosomes and microtubules simultaneously, during mitotic division in the early syncytial blastoderm stages. Using the objective lens in a water-dipping configuration, we applied our remote focusing to repeatedly acquire two-colour image volumes at a depth of approximately 5 µm under the

surface of the egg. We performed this imaging at a rate of 20 fps, equivalent to about 10 s per 3D volume, at room temperature for several minutes. The image stack of each volume required 210 raw images (5 phases × 3 angles × 2 channels × 7 sections) for the 3D-SIM reconstruction. We selected four time points within the first two minutes of the time series (*Figure 5A*), showing the synchronous transition from metaphase to telophase. Despite minor reconstruction artefacts (see Methods and materials for details), these results demonstrate that the remote focusing functionality of the system can be successfully applied for live 3D-SIM experiments, allowing four-dimensional acquisition while keeping the specimen stationary, thus avoiding the usual agitation and perturbations associated with mechanical actuation.

In the second example, where we applied remote focusing, we combined it with an advanced form of aberration correction. Thick biological samples, even adherent cells in culture, can induce optical aberrations of variable type and magnitude throughout the volume of the sample, owing to RI inhomogeneities. Aberration correction methods can be used to correct the aberrations in every imaging plane in the volume. However, this is a time-consuming process which can lead to unnecessary photobleaching and phototoxicity. A faster method is to perform the correction routine at fewer planes and then to estimate the correction modes for all other planes, an approach that we called 'multi-position aberration correction (MPAC).' We demonstrate MPAC in its simplest form, a linear estimation of two planes, which we found to work well, partially because some of the dominant aberrations, like spherical, are proportional to the imaging depth. By combining MPAC with remote focusing, we were able to demonstrate the full capability of Deep3DSIM's AO, as well as a large Z range of more than 16 μm (129 sections, 125 nm apart). We imaged fixed COS-7 cells in culture, stained for microtubules, to visualise the spindles during mitosis (*Figure 5B, C*). We measured the aberrations at two distinct positions, marked with arrows in *Figure 5B*, and then we estimated the corrections at different Z positions within the volume. Our results show conclusively that remote focusing with MPAC produced images with improved resolution compared to those without AO. Our method improved both the lateral and axial resolution (*Figure 5B*), and intensity profiles (*Figure 5C*). We conclude that the use of MPAC-based estimation, to correct aberrations for an entire imaging volume, and the use of remote focusing offer significant advantages to 3D-SIM imaging with an upright configuration.

## Discussion

We have demonstrated that the design of Deep3DSIM, a prototype upright super-resolution microscope with integrated AO, enables effective 3D super-resolution imaging within thick complex tissues to a depth of at least 130 μm. Its modular design and Python-based open-source software for both system control and GUI (see details in Methods and materials), make it comparatively user-friendly and versatile. These design concepts allow the adaptation of the instrument to suit specific research applications and enable protocols for experiments at scale. The upright optical design combined with aberration correction and remote focusing makes the instrument particularly suited for applications that require super-resolution volumetric imaging of live (or fixed) thick tissue, organs, or whole organisms. Deep3DSIM is also uniquely positioned for the imaging of the whole sample in expansion microscopy, without the need for mechanical sectioning, which is a challenge because of the softness and fragility of commonly used hydrogel techniques (*Cahoon et al., 2017*). Importantly, the design is compatible with specimen manipulation, such as microinjection, carrying out electrophysiology measurements, or liquid handling and exchange, while imaging experiments are in progress. In this study, we have established a new paradigm for 3D-SIM to be applied at depth while acquiring volumes without moving the specimen or the objective lens, and while varying the aberration corrections across the acquired volume, in a depth-specific manner. This general principle can now be applied to any SIM configuration, including an inverted one, which can make use of the highest possible NA objective lenses to obtain volumetric SIM datasets at depth with minimal artefacts and maximum possible spatial resolution.

Current commercially available 3D-SIM instruments are mostly based around inverted setups and high NA oil immersion objectives that are susceptible to aberrations when focusing into any specimens with RI mismatch. Such systems frequently only produce good images up to depths of a few μm. Silicone immersion objectives can improve the situation by allowing high-quality SIM up to a depth of 20 μm, despite having a slightly lower NA (*Ogushi et al., 2021*; *Richter et al., 2019*). In all cases, objective correction collars can only effectively minimise spherical aberrations for SIM imaging at a

single depth in a single channel. For multi-colour imaging, the RI of the mounting medium and the immersion medium, and the objective lens' correction collar setting, must be carefully tuned together to match the optimal wavelength-specific OTFs. Frequently, a compromise is made across different channels and imaging depths to accommodate multiple colours (*Demmerle et al., 2017*; *Wang et al., 2018*). Therefore, for most practical purposes, these microscopes can only effectively be applied to thin single layers of tissue culture cells on microscope slides or only to the superficial surface of complex tissues. Furthermore, they are not readily adapted to imaging regimes which require the specimen and the objective to remain motionless while imaging. Hence, Deep3DSIM fills an important application gap for deep imaging in live multi-cellular tissue, as well as fixed material. The method creates new possibilities for fast live super-resolution imaging while carrying out specimen manipulations. Such manipulations include micro-injection into *Drosophila* embryos and electrophysiology measurements on living mouse brain sections or *Drosophila* NMJ preparations.

An important consideration in 3D-SIM reconstruction is the precise settings of the algorithms, such as filtering. We chose to use a well-established reconstruction method based on Wiener filtering, as implemented in the softWoRx software suite, without advanced pre-processing of the image data for reduction of artefacts. This design decision enabled us to demonstrate the operating principles of Deep3DSIM and to adopt existing tools, such as SIMcheck for quality control of the raw and reconstructed data, and the overall characterisation of the imaging system. However, recently, several novel methods for 3D-SIM reconstruction have been developed (*Cai et al., 2022*; *Cao et al., 2023*). Such open-source software implementations should be compatible with Deep3DSIM, potentially achieving higher fidelity reconstructed data with reduced artefacts.

The resolution achieved with Deep3DSIM was 186 nm FWHM in the green channel using a water-based objective lens with an NA of 1.1. In comparison, inverted commercial 3D-SIM instruments routinely achieve ~120 nm FWHM resolution at the same wavelength using a high-NA oil immersion objective. While the Deep3DSIM resolution is, therefore, lower in absolute terms, this was a design trade-off chosen to make the microscope compatible with live imaging in an upright configuration and with a large working distance. Hence, it is more pertinent to compare the super-resolution of Deep3DSIM to that achievable for live cell imaging in widefield mode, at the same wavelength and with the same 1.1 NA water-based objective lens, which was 333 nm FWHM. Nevertheless, the fundamental design of Deep3DSIM could be readily adapted, if necessary, to work in upright or inverted mode with other high NA objective lenses. For example, oil or silicone immersion objective lenses, which would still fully benefit from the use of AO for aberration correction and motionless deep imaging. In these cases, the working distance would be compromised, making it much more difficult to manipulate the sample, for example, by microinjection or with electrophysiology tools. Although the design of Deep3DSIM was mostly optimised for upright microscopy with tissue specimens, it can also benefit other types of microscopy and applications. Notably, CryoSIM (*Kounatidis et al., 2020*) would hugely benefit from the inclusion of AO to correct for spherical aberrations induced by increased RI mismatch and instabilities at cryogenic temperatures, together with sample-induced aberration, particularly with non-flat overlying ice at the specimen-air interface.

Our AO methods provide only one average correction for the entire field of view (FoV). However, aberrations can differ across the FoV, and areas of higher image metric may bias the correction to better correct the aberrations contained within those areas. In practice, we found that field-dependent aberrations are usually not a problem for modest FoVs, such as the one in Deep3DSIM with a maximum diameter of about 96 μm. Nevertheless, we did experience residual aberrations when we imaged through the complex tissue of an entire *Drosophila* larval brain, approximately 130 μm deep, as measured with beads (*Figure 4B*). Zonal aberration correction can help with the compensation of field-dependent aberrations by splitting the FoV into multiple zones and then correcting for aberrations zone by zone, but it requires another level of instrumentation and control methods (*Rajaeipour et al., 2020*). Although the current Deep3DSIM design is not intended for this type of correction, it could be added to the instrument if required. Furthermore, the spatial light modulator (SLM), used for the generation of the SI, could also be used to compensate for field-dependent aberrations of the SI in the imaging plane (*Gong and Scherer, 2023*), in addition to the correction already provided by the DM.

The range of remote focusing shown in this work was limited to about ±8 μm, which is smaller than some previous demonstrations. There are two reasons why we did not use wider ranges in our

results. First, the DM can only reliably produce PSFs with minimal aberration repeatedly up to a range of about ±10 μm (see Appendix 2). Second, for 3D-SIM acquisition, ±5 μm is already a wide and commonly used Z range. Especially in multi-colour imaging, a rather large number of images, about 1200 frames (3 angles × 5 phases × 80 Z steps) per channel, would be required. While larger volume acquisition is possible, it would lead to considerable photobleaching and phototoxicity, as well as longer acquisition and reconstruction times. Nevertheless, a similar DM device has been used to achieve an impressive range of remote focusing (*Cui et al., 2021*). Although their approach is different from the high-NA super-resolution imaging presented here, a stable DM should increase the practical range of our remote focusing approach twofold or even greater. In terms of speed, refocusing with the DM instead of using the piezo stage reduced the duration of the image acquisition significantly, because the settling time of the surface of the DM was around an order of magnitude faster than the settling time of the piezo stage, which highlights the potential of AO-based refocusing for fast imaging (see Appendix 3 for further details).

There are various routes through which the system performance could be enhanced in future versions of Deep3DSIM. First, the speed of acquisition in the reported demonstration was limited by a few key system components, which could be improved in future developments. A key limiting factor in widefield mode was the EM-CCD camera. The best overall performance in terms of SI modulation was achieved when using conventional CCD modes. However, due to the data transfer time (3 MHz pixel readout rate), the frame rate was limited to about 100 fps for a reasonably sized FoV. This acquisition speed could be increased by using modern sCMOS cameras. In addition, in SIM experiments, the imaging speed was also limited by the update rate of the SLM, determined by the settling time of the nematic liquid crystal technology (>20 ms), giving a maximum frame rate of 50 fps raw data. Fast SLMs using ferroelectric liquid crystal materials have been demonstrated in SIM for high-speed SIM imaging (*Lin et al., 2021*), though it is more challenging to use such SLMs for multi-colour imaging, as the phase modulation is binary and hence can only be optimised for discrete wavelengths. Third, future improvements could also include the incorporation of denoising algorithms into the reconstruction process to allow lower light dosage, which would enable faster imaging, reduced photobleaching, and reduced specimen photodamage (*Huang et al., 2018*; *Smith et al., 2021*). Finally, the Deep3DSIM prototype was custom-built on a large optical table, instead of basing the design around a commercial upright microscope stand, because the latter would have imposed unnecessary restrictions at this prototyping stage. In future, the optical path of Deep3DSIM could be simplified and reduced in size. Such a future design would make it easier to adopt or commercialise and to modify Deep3DSIM for bespoke applications.

## Materials and methods

### Microscope

We based the Deep3DSIM system on our previous system, Cryo-SIM (*Phillips et al., 2020*), which was developed from earlier work in John Sedat's group at UCSF (*Dobbie et al., 2011*; *Gustafsson et al., 2008*). The microscope body was built in an upright configuration (*Figure 1*) around a bespoke-engineered scaffolding arch, which we designed to ensure sufficient mechanical stability for super-resolution microscopy while accommodating a pair of micromanipulators for electrophysiological experiments. We used a water-immersion objective lens (Olympus LUMFLN60XW), capable of working in a water-dipping configuration by adjusting a correction collar, because its combination of high NA and long working distance made it most suitable for deep imaging of tissues, particularly *Drosophila* brain and NMJ, and for electrophysiological experiments. More details about the optical setup can be found in Appendix 4.

### Structured illumination

We used a nematic liquid-crystal SLM (Boulder Nonlinear Systems HSP512-488-800) to create SI with optimal contrast on the sample (modulation contrast). The SLM was positioned in the sample-conjugated plane and linear sinusoidal patterns were generated for three orientation angles. The ±1st order and zero-order beams were used to interfere at the sample plane to generate the 3D SI. To achieve optimal interference contrast, the linear polarisation of the excitation beam from the SLM was rotated using a liquid crystal device (Meadowlarks Optics LPR-100-$\lambda$) to maintain S polarisation

at the incident plane at the back focal plane (BFP) of the objective lens for each SI orientation. For widefield acquisitions, the SLM was switched off and used as a single reflector so that only the zero-order beam provided epi-illumination, while for SIM acquisition, the SLM was switched on and it was preloaded with the patterns for the specific wavelength and line width. During image acquisition, the SI phase and angle were rotated in a fixed sequence, synchronised with the polarisation rotators, the light sources, and the cameras.

## Deformable mirror

We used a DM (Alpao DM69-15) which was positioned between the main dichroic filter and the microscope body, conjugated to the BFP of the objective lens. The DM was used for the correction of aberrations in both the excitation and the emission paths. The DM aperture was matched to the BFP by an optical relay system. We included a pair of flip mirrors to create an optical path that bypassed the DM (dotted path in *Appendix 4—figure 1*), allowing easy comparison between imaging without and with AO.

## Software

The system control software package was composed of three parts (*Figure 1C*): *python-microscope* for low-level hardware control of the microscope system (*Susano Pinto et al., 2021*), *microscope-cockpit* for a user interface and additional functionality such as experiments (*Phillips et al., 2021*), and *microscope-aotools* for performing DM calibration and AO aberration correction (*Hall et al., 2020*). All three were further developed and improved to match the requirements of Deep3DSIM. Most notably, remote focusing was added to the *microscope-aotools* package. All changes were tracked and they are described in more detail in the respective branch repositories:

- *python-microscope*: https://github.com/dstoychev/microscope/tree/deepsim
- *microscope-cockpit*: https://github.com/dstoychev/cockpit/tree/deepsim
- *microscope-aotools:* https://github.com/dstoychev/microscope-aotools/tree/deepsim-matthew-danny

The complex orchestration and timing of experiments, carried out by Cockpit, was implemented with a specialised embedded computer (Red Pitaya STEMlab 125–14 v1.0).

For SIM data reconstruction, we used the commercial software softWoRx (Cytiva), which uses Wiener deconvolution, based on the approach by *Gustafsson et al., 2008*. OTFs required for reconstruction for each imaging channel were generated from experimental data acquired from a single fluorescent bead.

Multi-colour image alignments were performed in the open-source software Chromagnon (*Matsuda et al., 2020*). SIM characterisation was done with SIMcheck (*Ball et al., 2015*). General image processing was done in Fiji (*Schindelin et al., 2012*).

## Sensorless AO aberration correction

We used standard Zernike-based modal sensorless AO, as described previously (*Antonello et al., 2020*; *Hall, 2020*). We calibrated the DM with an integrated interferometer (*Appendix 4—figure 1*, dashed path) to obtain an accurate Zernike mode control matrix, which we used for both aberration correction and remote focusing. The duration of the calibration routine was around 30–60 min, mostly spent on phase unwrapping and other intensive computations. The resulting control matrix could be reused for weeks at a time, before recalibration was necessary, and potentially much longer on devices which are not subject to creep effect.

Our base set of correction modes included 8 Zernike modes: #5 (primary oblique astigmatism) to #11 (primary spherical) and #22 (secondary spherical), following Noll indexing. Example correction modes and their amplitudes for each sample are given in the figure supplements of *Figure 3* to *Figure 5*. Our aberration correction method used a standard optimisation algorithm where each mode was scanned individually by applying several different mode biases to the DM, e.g., $\{-1, -0.5, 0, 0.5, 1\}$ rad, and acquiring widefield images each time. For our base set of correction modes, this scanning arrangement resulted in 40 images in total. Metric values were calculated for each image and a Gaussian curve was fitted to the amplitude-metric points. The mean of the Gaussian was used as the optimal correction for the Zernike mode. We used a Fourier-based metric which is described in Appendix 5. We observed that ISOsense (*Žurauskas et al., 2019*) achieved the most robust correction (data not

shown) when the images did not have prominent high spatial frequency content, i.e., few or no sub-diffraction features.

In our approach to aberration correction, we always corrected the system-induced aberrations first, by using a sparse bead sample and by tuning the objective lens collar. Any subsequent correction of sample-induced aberrations was done on top of this 'system flat' correction. When imaging, we always carried out the aberration correction routine first and then held the DM in the corrective shape during the actual image acquisition. For relatively thin volumes, e.g., 1–2 µm thickness, we mostly corrected just a single plane (usually the middle of the stack), but for thicker volumes, we developed our MPAC approach. The duration of the correction routine for a single plane was proportional to the number of modes involved, the number of scanning points for each mode, and the camera exposure configuration, but usually in the order of seconds. The earlier example of 40 images for our base set of corrections would normally take around 10 s (less than 4 s of actual light exposure). The correction was independent of the subsequent image acquisition. This independence between the two processes let us configure separately the exposure settings for the correction routine, allowing us to tune parameters, such as exposure time and laser power, in a way that minimises issues such as low contrast and noise. We further accounted for noise in our Fourier-based image metric, as described in Appendix 5.

## Multi-position aberration correction

We used MPAC to compensate for the optical distortions in thick volumes. We applied our standard sensorless AO correction routine at fixed positions, e.g., at the top and the bottom of the volume, and then we did a linear fitting of the resulting correction Zernike modes. This simple approach allowed us to calculate in advance the corrections for every section of the volume. All corrections required for an imaging session were stored in the DM driver beforehand and then rapidly iterated via electrical (TTL) signals in real-time during the session, same as the remote focusing function.

## Remote focusing

We controlled the remote focusing and the aberration correction independently, each one with its own set of corrections, which were then summed together if both functionalities were to be used simultaneously. To calculate the required remote focusing DM patterns, we first performed a calibration procedure, which was based on our aberration correction method and hence used the same type of optimisation algorithm. Our approach achieved a linear response with high precision, within a range of ±5 µm. Because of instability problem with the DM, we then had to perform a second calibration step to achieve high accuracy as well. We describe all this tuning in Appendix 2, and we elaborate on the instability problem in Appendix 6. Similarly to the MPAC, all remote focusing patterns were computed in advance before the image acquisition and then they were iterated by TTL triggers. When remote focusing was combined with 3D-SIM imaging, the focal plane and the SI pattern were synchronously moved together during axial scanning, keeping constant the phase relationship between the two. The change in effective NA due to focus shift was negligible with our choice of focusing range (up to ~16 µm) and objective lens.

## SIM reconstruction artefacts

Super-resolved images in 3D-SIM are obtained with a reconstruction process which can sometimes lead to artefacts (*Demmerle et al., 2017*). In this study, we intentionally chose difficult-to-image samples, to demonstrate the full capability of the Deep3DSIM prototype and to show realistic examples.

Optical aberrations are a major source of reconstruction artefacts, and this is very well illustrated in *Figure 3A*. Without AO, the weak signal in the green channel resulted in high spatial frequency noise (hammerstroke) and stripe (hatching) artefacts. Furthermore, the enhanced RI mismatch led to increased spherical aberrations, and subsequently to ghosting artefacts, manifesting as faint ripples in some places in the red channel (arrow in *Figure 3A*). As expected, all these artefacts were removed in the AO corrected images (*Figure 3B*).

However, in some situations, AO correction cannot completely remove all artefacts. We demonstrate this in *Figure 4B*, where the corrected volumes still have visible artefacts. There are two main reasons for this. First, high levels of optical scattering are experienced at such extreme imaging depth without optical clearing of the tissue. The scattering is a problem not only on the excitation side, leading to poor modulation contrast of the SI, but also on the emission side where it leads to weak

fluorescence signal with poor contrast. These problems resulted in hatching artefacts, most noticeable in the green channel (*Figure 4B*, right). On the other hand, an extreme imaging depth also leads to large and complex aberrations which may not be entirely corrected. For example, some of the aberrations may have a magnitude that cannot be adequately compensated by the wavefront corrector, such as a DM, or likewise, they may contain high-order components which cannot be faithfully reproduced. In our case, such residual spherical aberrations resulted in ghosting artefacts in both channels (*Figure 4B*, right).

Finally, we also observed pronounced hatching artefacts in the green channel of our live imaging of a *Drosophila* embryo (*Figure 5A*), which was caused by poor modulation contrast at one of the SI angles.

## Cell culture

COS-7 (ATCC, CRL-1651) cells were grown on #1.5 glass cover slips in DMEM containing 10% foetal bovine serum (Sigma-Aldrich), washed twice with 2 ml of 100 mM sodium phosphate and fixed for 2 h in 2 ml PLP (2% [wt/vol] paraformaldehyde in 87.5 mM lysine, 87.5 mM sodium phosphate at pH 7.4, and 10 mM sodium periodate). Cover slips were washed three times in 2 ml (100 mM) sodium phosphate, pH 7.4, before permeabilisation in 1 mg/ml BSA, 0.05% saponin, and 100 mM sodium phosphate, pH 7.4, for 30 min. In all cases, primary antibody (α-Tubulin (mouse; DM1A Santa Cruz) AF 488 and Rtn4 (rabbit; AbD Serotec) AF 555) staining was performed in 1 mg/ml BSA, 0.05% saponin, and 100 mM sodium phosphate, pH 7.4 for 60 min at room temperature. Affinity-purified antibodies were used at 1 µg/ml; commercial antibodies were used as directed by the manufacturers. DAPI was added to the secondary antibody staining solution at 0.3 µg/ml. Cover slips were mounted in Mowiol 4–88 mounting medium (EMD Millipore).

The microtubules stained in this way occasionally appeared fragmented, more noticeable in the SIM images in *Figure 2B*. This fragmentation is normal for a sample preparation protocol, such as ours, based on 2% paraformaldehyde fixation. More advanced protocols, specifically designed for the preservation of structures such as microtubules, can result in more continuous filaments.

## Fly strains

The following fly lines were used in this study. Nrx-IV::GFP (CA06597, kindly gifted by the Rita Teodoro lab) for the NMJ samples, cno::YFP (CPTI000590, DGRC 115111) for the brain samples, Jupiter::GFP, his2A::RFP (*Hailstone et al., 2020*) for embryos. All stocks were raised on standard cornmeal-based medium at 25°C.

## Fixed *Drosophila* larval fillet for imaging of neuromuscular junctions

The NMJ samples were prepared by following the protocol in *Brent et al., 2009*, apart from the fixation, which was done in 4% paraformaldehyde (PFA) in PBSTX (phosphate-buffered saline with 0.1% Triton X-100) for 30 min and then followed by two washes for 20 min each in PBSTX. The samples were then incubated with Cy3-conjugated α-HRP antibody (1:500, Jackson ImmunoResearch 123-165-021) for 1 hr. Finally, the samples were washed in PBSTX for 45 min and then mounted in Vectashield. All steps were done at room temperature.

## Fixed whole *Drosophila* brains

Third instar *Drosophila* larvae were dissected in PBS. Brains were removed and fixed in 4% PFA in PBSTX (0.3% Triton X-100) for 25 min. Afterwards, they were rinsed three times in PBSTX, further permeabilised with two 20 min washes, and then mounted in PBS.

## *Drosophila* embryos for live imaging

The live embryo samples were prepared as described in *Parton et al., 2010*, except that they were mounted in PBS.

## Acknowledgements

John Sedat for the original optical design; Antonia Göhler and Mantas Žurauskas for the initial optical characterisation, work which was published previously; Martin Hailstone, Francesca Robertson, and Jeff Lee for preparing specimens during various phases of using the Deep3DSIM instruments with

biology, imaging that was not shown in the manuscript. JW thanks Jacopo Antonello, Chao He, and Jiahe Cui for insightful discussions and advice for AO devices. We are grateful to Micron Oxford and its numerous partners and staff for discussions and providing the environment required for the success of this complex interdisciplinary technology development project.

## Additional information

### Funding

| Funder | Grant reference number | Author |
|---|---|---|
| Wellcome Trust | 10.35802/091911 | Mick A Phillips<br>David Miguel Susano Pinto<br>Richard M Parton<br>Nicholas James Hall<br>Ian Dobbie<br>Ilan Davis |
| Medical Research Council | [MR/K01577X/1] | Ilan Davis |
| Wellcome Trust | 10.35802/096144 | Mick A Phillips<br>David Miguel Susano Pinto<br>Richard M Parton<br>Nicholas James Hall<br>Ian Dobbie<br>Ilan Davis |
| Wellcome Trust | 10.35802/105605 | Mick A Phillips<br>David Miguel Susano Pinto<br>Richard M Parton<br>Nicholas James Hall<br>Ian Dobbie<br>Ilan Davis |
| Wellcome Trust | 10.35802/107457 | Mick A Phillips<br>David Miguel Susano Pinto<br>Richard M Parton<br>Nicholas James Hall<br>Ian Dobbie<br>Ilan Davis |
| Wellcome Trust | 10.35802/203141 | Mick A Phillips<br>David Miguel Susano Pinto<br>Richard M Parton<br>Nicholas James Hall<br>Ian Dobbie<br>Ilan Davis |
| Wellcome Trust | 10.35802/209412 | Mick A Phillips<br>David Miguel Susano Pinto<br>Richard M Parton<br>Nicholas James Hall<br>Ian Dobbie<br>Ilan Davis |
| European Research Council | 10.3030/695140 | Jingyu Wang<br>Matthew Wincott<br>Martin J Booth |
| HORIZON EUROPE Marie Sklodowska-Curie Actions | 10.3030/766181 | Lothar Schermelleh |
| Biotechnology and Biological Sciences Research Council | [BB/M011224/1] | Danail Stoychev |

The funders had no role in study design, data collection and interpretation, or the decision to submit the work for publication. For the purpose of Open Access, the authors have applied a CC BY public copyright license to any Author Accepted Manuscript version arising from this submission.

## Author contributions

Jingyu Wang, Development of control software; developed adaptive optics methods, algorithms and calibration; Implemented remote focusing method; Refined and reimplemented optical system and adaptive optics control; Refined experimental processes; Experimental strategy and specimen preparation; Imaging experiments, calibration, and data processing; Article conceptualisation; Initial draft of article and figure preparation; Revision of article; Danail Stoychev, Implemented electronics; Development of control software; Developed adaptive optics methods, algorithms and calibration; Implemented remote focusing method; Refined and reimplemented optical system and adaptive optics control; Refined experimental processes; Experimental strategy and specimen preparation; Imaging experiments, calibration, and data processing; Article conceptualisation; Initial draft of article and figure preparation; Revision of article; Mick A Phillips, Designed the microscope optics; Designed and implemented the optomechanics; Implemented electronics; Implemented initial optical system; Development of control software; David Miguel Susano Pinto, Development of control software; Richard M Parton, Original vision and concept development; Acquisition of initial data; Refined experimental processes; Experimental strategy and specimen preparation; Article conceptualisation; Nicholas James Hall, Implemented initial optical system; Development of control software; Developed adaptive optics methods, algorithms and calibration; Acquisition of initial data; Joshua S Titlow, Acquisition of initial data; Ana Rita Faria, Refined experimental processes; Experimental strategy and specimen preparation; Matthew Wincott, Development of control software; Dalia Gala, Experimental strategy and specimen preparation; Andreas Gerondopoulos, Experimental strategy and specimen preparation; Niloufer Irani, Experimental strategy and specimen preparation; Ian Dobbie, Original vision and concept development; Designed the microscope optics; Implemented electronics; Implemented initial optical system; Development of control software; Acquisition of initial data; Revision of article; Supervision; Lothar Schermelleh, Refined experimental processes; Experimental strategy and specimen preparation; Imaging experiments, calibration, and data processing; Article conceptualisation; Revision of article; Obtaining funding; Supervision; Martin J Booth, Developed adaptive optics methods, algorithms and calibration; Article conceptualisation; Revision of article; Obtaining funding; Supervision; Ilan Davis, Original vision and concept development; Designed the microscope optics; Article conceptualisation; Initial draft of article and figure preparation; Revision of article; Obtaining funding; Supervision

## Author ORCIDs

Jingyu Wang ⓘ https://orcid.org/0000-0002-2856-7602
Danail Stoychev ⓘ http://orcid.org/0000-0001-5539-2206
Mick A Phillips ⓘ https://orcid.org/0000-0003-3578-7301
Joshua S Titlow ⓘ https://orcid.org/0000-0002-9586-4797
Ian Dobbie ⓘ http://orcid.org/0000-0002-5531-5865
Lothar Schermelleh ⓘ https://orcid.org/0000-0002-1612-9699
Martin J Booth ⓘ https://orcid.org/0000-0002-9525-8981
Ilan Davis ⓘ https://orcid.org/0000-0002-5385-3053

## Decision letter and Author response

Decision letter https://doi.org/10.7554/eLife.102144.sa1
Author response https://doi.org/10.7554/eLife.102144.sa2

# Additional files

## Supplementary files
MDAR checklist

## Data availability
The Python software used to control the microscope is available on GitHub: python-microscope (copy archived at *Stoychev, 2022a*); microscope-cockpit (copy archived at *Stoychev, 2022b*); microscope-aotools (copy archived at *Stoychev, 2023*).

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

## Appendix 1

### Structured illumination pattern generation

The stripe width of the sinusoidal SI was 317 nm for the green channel and 367 for the red channel. We used a SLM to diffract the incoming excitation beam into three paths, consisting of the zero-order beam and the ±1 order diffraction beams. The three beams interfered at the sample plane to create an axially varying stripe pattern. An example of the pattern is shown in *Appendix 1—figure 1*.

The width of the stripe pattern was chosen to be larger than the theoretical possible, e.g., 271 nm in the green channel by using the Rayleigh criterion for resolution. There were three reasons for this choice. First, in the frequency domain, we observed that the highest spatial frequency amplitudes decreased as the resolution approached this theoretical limit. Using a slightly larger stripe width drastically improved the intensity of these amplitudes, resulting in improved SNR of the high-resolution features in the reconstructed images. Second, the radial distance of the first-order beams in the BFP, and hence on the DM aperture, increases as the resolution decreases. However, the actuation of the reflective membrane of the DM was less effective at the periphery because of boundary conditions. Third, higher spatial frequency stripes meant a higher incidence angle of the outer beams on the sample and the effect of spatially varying aberrations was further enhanced at these higher angles. The change in angle could be corrected in principle, because the beams were focused on the DM, but the field-dependent aberrations that affected each individual beam differently could not be fully corrected in this way.

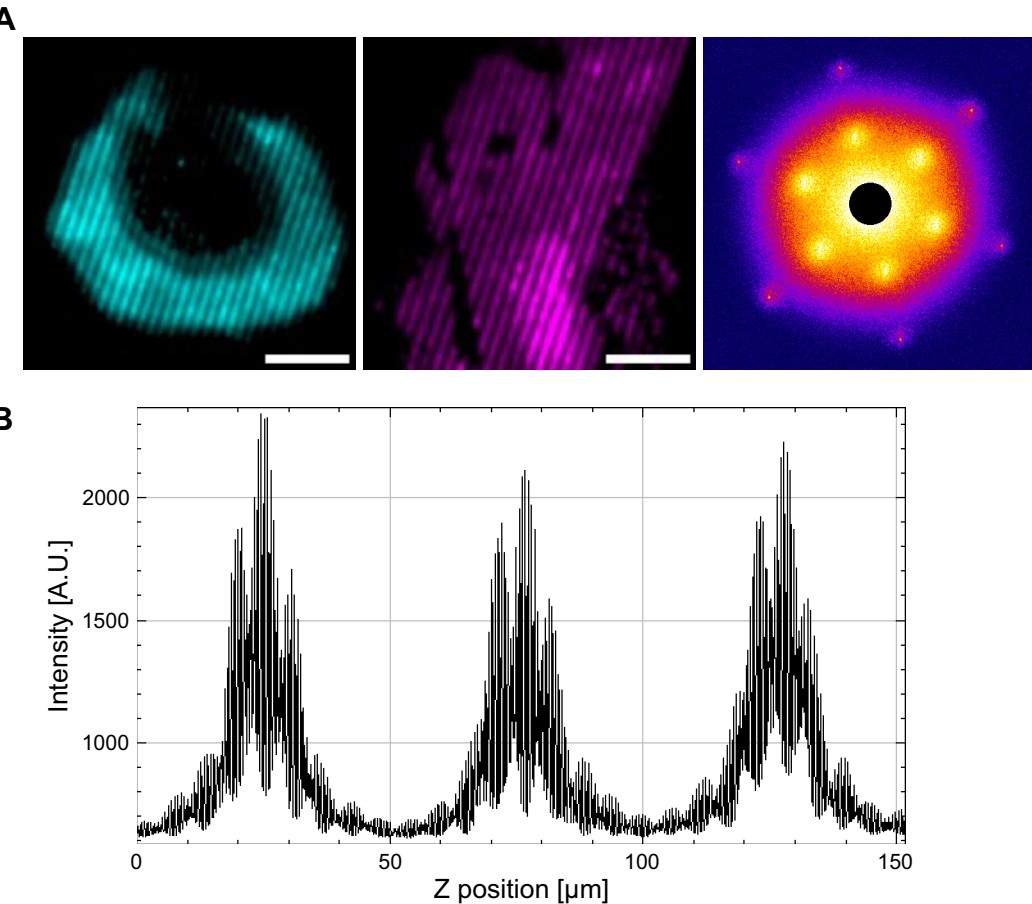

**Appendix 1—figure 1.** Structured illumination microscopy (SIM) pattern generation. (**A**) Example of the structured illumination (SI) pattern at one of the angle orientations and phase shifts, visualised by imaging a monolayer of 100 nm diameter fluorescent beads in the green (left) and in the red (middle) channels. The power spectrum (right) shows combined frequency response in all three angles of the green channel, with the centre masked with a black

*Appendix 1—figure 1 continued on next page*

*Appendix 1—figure 1 continued*

circle to create a better contrast. Scale bars: 5 µm. (**B**) Fluorescence signal modulated with structured illumination, taken from the centre of a single red fluorescent bead. The plot shows three 50 µm scans through the bead, one for each of the SI angles.

## Appendix 2

### Remote focusing calibration and range

We implemented the remote focusing by reshaping the wavefront phase with the DM, with same type of Zernike-based modal control as the aberration correction. Effectively, it was a mapping between Zernike mode coefficients and Z positions. We created this mapping with a sample of a single layer of sparse 100 nm fluorescent beads; we first introduced an axial displacement with the piezo stage and then we used the DM to bring the beads back into focus with our sensorless aberration correction by also including the defocus mode.

We validated the remote focusing by displacing the beads sample by four offsets, $\{-5, -2.5, 2.5, 5\}$ μm, and acquiring a Z-stack for each offset. We then checked if the displacements of the beads within the Z-stack matched the displacement of the stage. We repeated the same procedure with just the piezo stage, using the data as a ground truth. Each volume was acquired 10 times for statistical analysis, and likewise, we analysed the same 10 beads from each volume. We used Gaussian fitting to find the centre Z position of each bead. In terms of precision, we calculated the exact pooled variance of each of the offsets to be $\{6.45, 4.15, 20.58, 11.83\}$ nm, respectively. This demonstrated the good repeatability of the DM refocusing. However, the images showed mismatches in the position measurement (*Appendix 2—figure 1A*) and the image of a single bead was elongated (*Appendix 2—figure 1B*). We interpreted this to be caused by the creep effect of the DM. The findings from Appendix 6 demonstrated two important observations: (1) most of the initial change in wavefront phase due to the creep effect was within the first minute; (2) the exponential rate of change meant that the creep effect was effectively linear during this initial period. The rate and the way in which the shape of the DM was changed during the calibration of the remote focusing was different from those used during subsequent image acquisitions. Therefore, the calibration compensated for more creep than was typically encountered during actual imaging experiments. However, the rates of both the calibration and the imaging were well below the 1 min threshold, typically in the order of milliseconds, and therefore the contribution of the creep effect was only different up to a scale, because of the linearity in this region. We calculated this scaling factor as the ratio of the slopes of the lines in *Appendix 2—figure 1A*, in this case yielding a value of around 0.7. This scaling approach allowed us to get accurate remote Z displacements that matched the ground truth displacements obtained with the piezo stage, as shown in *Appendix 2—figure 1C*.

We usually performed remote focusing with a range of ±5 μm or less, because this is already a large volume, requiring thousands of images to be acquired for multi-colour 3D-SIM, and because the image quality was optimal in this range. We sometimes extended this range, e.g., to the ±8 μm range shown in *Figure 5B*, but we noticed that the quality of the PSF decreased beyond the ±10 μm range. We attribute this to the larger actuator control voltages, applied over longer periods of time, which meant that the impact of the creep effect on the DM was even stronger.

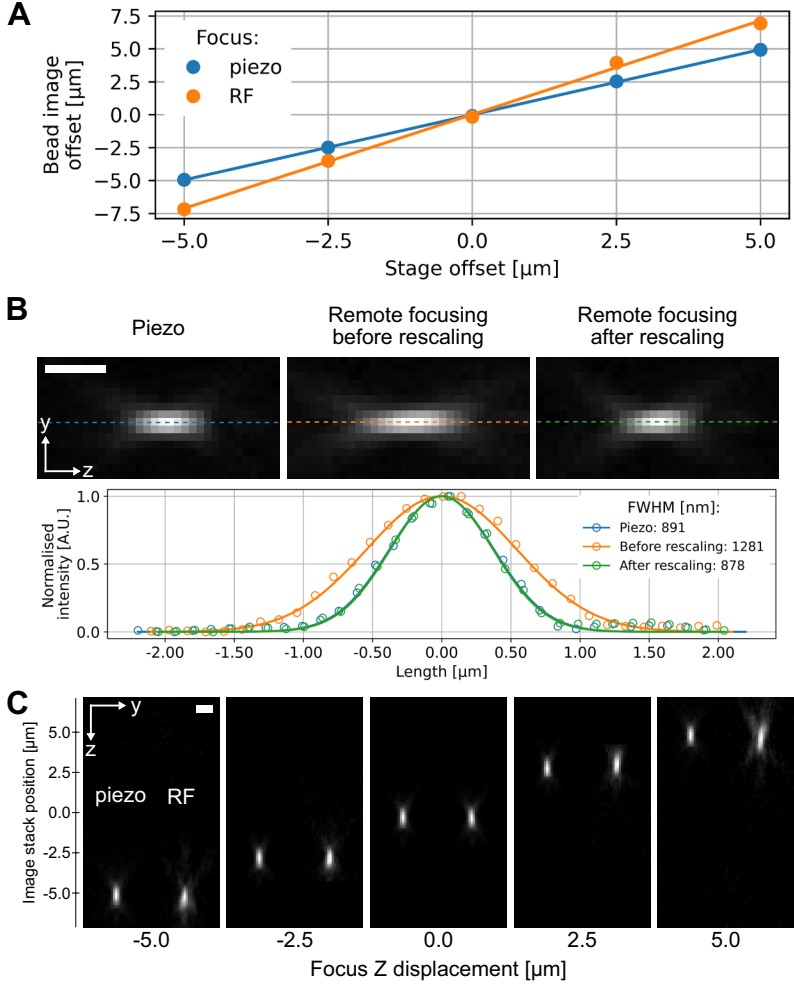

**Appendix 2—figure 1.** Effect of rescaling on remote focusing. (**A**) Comparison of remote focusing to conventional refocusing with piezo stage. (**B**) Demonstration of how pixel rescaling restores the axial profile of bead images acquired with remote focusing. (**C**) Displacement comparison between piezo (left) and remote focusing (right). Scale bars: 1 μm.

## Appendix 3

### Axial scanning speed

The remote focusing was desired not only for its ability to scan the sample axially while keeping it stationary, but also for the faster scanning speed compared to piezo stages. In our case, this was already indicated from the manufacturers' datasheets, which listed the settling times of the DM and the Piezo as 0.4 ms and 7 ms (10% step width), respectively. The settling time is the time it takes for the device to reach a new position and maintain it without oscillation. In the case of the piezo stage, this happens every time the device moves up or down, whereas for the DM, it is a change in the shape of the reflective surface. Clever execution of imaging experiments can negate a lot of this 'dead time' by, for example, reading out image data from the detectors in the meantime (Cockpit already does this). However, fast image acquisition with low readout times can lead to a situation where the settling time needs to be waited out. We measured the step responses of the DM and the piezo stage, which allowed us to estimate the settling times. The values we obtained were substantially higher than the reported values from the respective datasheets. The response of the DM was measured by checking the fringe patterns of the integrated interferometer. Limited by the interferometer camera readout speed, we estimated the settling time to be between 1 and 2 ms. The response of the piezo stage was measured both with digital (using the manufacturer's software PIMikroMove) and with analogue (using an oscilloscope) methods, with different step sizes. Using default PID parameters, we found the 10% step width setting time to be around 25 ms. However, for experiments, we used a step of size 125 nm, for which the settling time was around 10ms. In conclusion, the settling time of DM was close to an order of magnitude faster than the piezo stage.

# Appendix 4

## Optical design

An optical diagram of Deep3DSIM is shown in *Appendix 4—figure 1*. The laser module was composed of four commonly used laser lines: 405, 488, 561, and 642 nm. The 405 nm laser was incompatible with the SLM and thus was only used in widefield mode. We ended up doing all experiments with the 488 nm and 561 nm channels, so there are only two detectors in the diagram, but two more can be added for 4-channel simultaneous imaging. The system could easily be scaled to use more than four channels. By using a half-wave plate in each path, all laser beams had their polarisation rotated to be parallel to the slow axis of the SLM (Boulder Nonlinear Systems HSP512-488-800). The four beams were then combined by mirrors and dichroic filters into the same path before being guided to the SLM. The flip mirrors FM5 and FM6, together with the FBS beamsplitter, were used to bypass the SLM and create an integrated interferometer (dashed path) for the calibration of the DM. The excitation beam was deflected by the SLM towards the DM. By applying a sinusoidal stripe pattern to the SLM at a sample conjugated plane, the excitation path was diffracted into three beams consisting of $\pm 1^{st}$ order and zero-order beams. The three beams interfered at the sample plane below the objective lens to form the 3D SI. The foci of the three beams were filtered by a 7-point spatial filter (A2). A polarisation rotator (Meadowlarks Optics LPR-100) was used to change the linear polarisation angle of the structured illumination pattern during experiments. The excitation light and the emission light were separated by a dichroic filter (D4, Chroma Technology ZT405/488/561/640rpc) that was aligned at 22.5° from the optical axis to achieve the optimal sharpness of the transitions between the transmission (excitation) and reflection (emission) for all wavelengths. The combined excitation beam was deflected by a DM (Alpao DM69-15) which was conjugated to the BFP of the 60× objective lens (Olympus LUMFLN60XW). A 10× air objective lens (Olympus LMPLFLN10X) was used for widefield imaging with a large FoV (dash-dotted path). Flip mirror FM2 was used to switch between the two objective lenses, whereas flip mirror FM1 was used to direct the white LED light LL to one of the objective lenses. Note that only the 60× objective lens was optimised and intended for SIM and AO. The pair of flip mirrors FM3 and FM4 was used to bypass the DM (dotted path) for comparison between AO and no AO. The sample was moved in 3D with an assembly of two motorised piezo stages, PI M-687 (XY) and PI P-736 (Z). This entire assembly was further mounted on a heavy-duty Z-stage (Aerotech PRO115), which was used for coarse large-range Z control. The emission light was collected simultaneously from multiple channels by EMCCD cameras (Andor iXon Ultra 897). The separation of green and red fluorescence was done with a dichroic filter (D5, Chroma ZT561rdc-xr), and each channel was further filtered by individual band-pass filters: ET525/50 m (Chroma) for the green channel and ET600/50 m (Chroma) for the red channel.

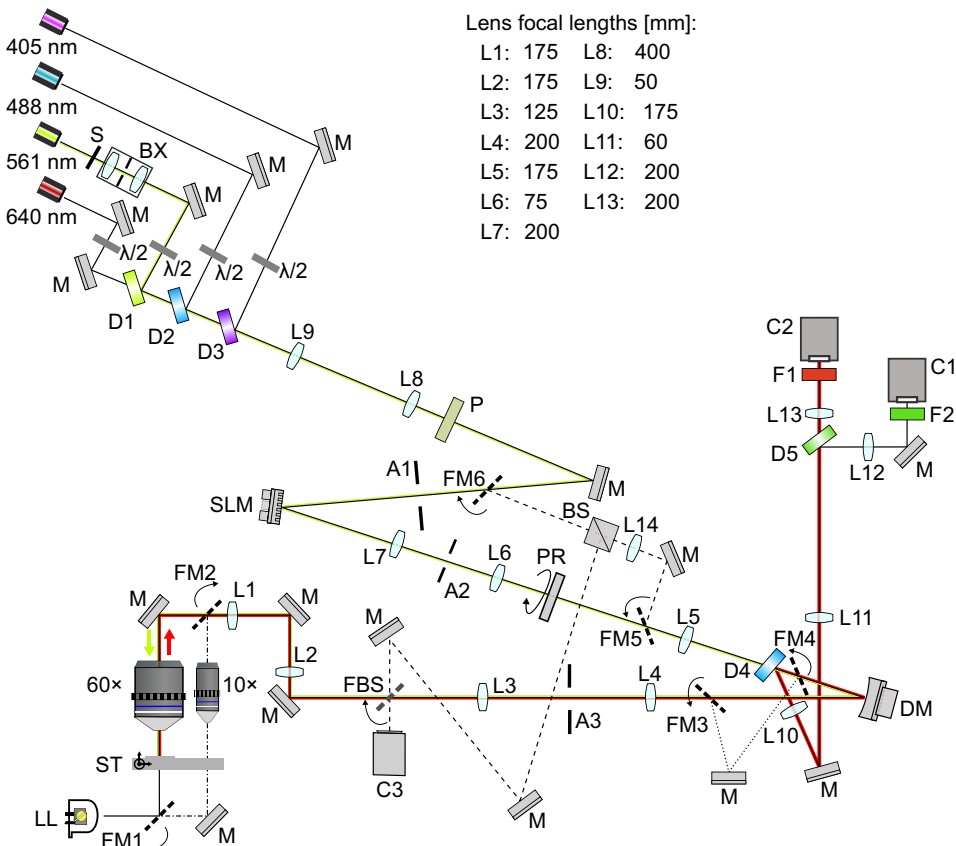

**Appendix 4—figure 1.** Optical arrangement. 405 nm, 488 nm, 561 nm, and 640 nm: laser sources. S: mechanical shutter. BX: beam expander. F1 to F2: optical filters. M: mirrors. $\lambda/2$: half-wave plates. L1 to L13: lenses, whose focal lengths are given in the table at the top of the figure. P: polariser. SLM: spatial light modulator. A1 to A3: apertures. FM1 to FM6: motorised flip mirrors. PR: polarisation rotator. BS: beamsplitter. FBS: 50/50 beamsplitter on flip mount. D1 to D5: dichroic beamsplitters. DM: deformable mirror. C1 to C2 are EMCCD cameras, and C3 is a CCD camera for the wavefront sensing interferometer. 10× and 60× are objective lenses. ST: assembly of stages (X, Y, and Z movement). LL: LED light used for brightfield imaging.

## Appendix 5

### Image metric

We used an image metric based on Fourier-domain analysis to quantify the image quality during the aberration correction routine. The algorithm, as well as other metrics, was previously developed as part of the *microscope-aotools* software package. An image $I(x, y)$ was transformed to frequency space and a logarithmic power spectrum $S(u, v)$ was obtained:

$$S(u, v) = \ln\left(\left|\mathfrak{F}\{I(x, y)\}\right|^2\right)$$

The zero-frequency component was shifted to the centre of the spectrum. The Rayleigh criterion was used to derive the cutoff frequency $\omega_c$, which was used to define two filters, one high-pass and one band-pass:

$$F_H = \begin{cases} 1, & \text{if } \|(u, v)\|_2 > 1.1\omega_c \\ 0, & \text{otherwise} \end{cases}$$

$$F_B = \begin{cases} 1, & \text{if } \omega_c > \|(u, v)\|_2 > 0.1\omega_c \\ 0, & \text{otherwise} \end{cases}$$

The high-pass filter was used to separate the high-frequency noise and then to define a noise threshold $\bar{S}_n$ as the mean of all noise pixels:

$$\bar{S}_n = \frac{1}{UV} \sum_u^U \sum_v^V S(u, v) \cdot F_H$$

The band-pass filter was used to further remove the contributions of the 0-frequency and low-frequency regions, often marked by strong artefacts (e.g. border effect) and containing signal of little interest, such as background light and blur. This created the final binary set of pixels associated with the metric:

$$S_m = \begin{cases} 1, & \text{if } \left(S(u, v) \cdot F_B\right) > \bar{S}_n \\ 0, & \text{otherwise} \end{cases}$$

Finally, the numeric value of the metric was derived by simply counting the number of non-zero pixels in the subset:

$$\left|\{s \mid s \in S_m(u, v) \text{ and } s = 1\}\right|$$

In summary, the metric quantified the spread of the spatial frequency content in the image, and the metric was at its maximum in the absence of aberrations.

# Appendix 6

## DM creep effect

We observed that our DM (Alpao DM69-15) exhibited certain instability, where the surface of the mirror deviated from its intended shape over time. We note that this particular DM was a model manufactured in 2014; we note also that, to our knowledge, later revisions of the same model have been improved to remove such effects. Unlike the well-known hysteresis, this temporal behaviour known as 'creep' has been reported in the literature, e.g., (*Bitenc, 2017*; *Bitenc et al., 2014*), and it has been attributed to a combination of drift and temperature effects. To better understand the dynamic behaviour of the device, we performed a series of experiments.

The first experiment aimed to test the long-term stability of the device. The total duration of the experiment was 16 hr, and it was divided into 1 min cycles. For the first 50 s of each cycle, random offsets were added to the shape, with a frequency of several hundred hertz. The idea was to simulate normal operation of the device. The offsets were uniformly distributed in the range ±5% of the total range, so on average, the shape remained the same. In the final 10 s of each cycle, the original shape was re-applied and kept for the remainder of the cycle. Interferograms were captured at the end of each cycle, and root mean square (RMS) wavefront phase error from the original shape was calculated (*Appendix 6—figure 1A*). This experiment confirmed our observation about the change in shape and it showed significant changes within the first hour, as well as a steady increase in error even after 16 hr.

To further distinguish between the effects of drift and temperature, we performed a second set of experiments. The device was warmed up and initialised by holding defocus for 4 hr. Six different shapes were then applied and monitored for 20 min each. Then a final shape was held for 60 min. In all these experiments, the DM was undisturbed until the next change in shape. All the shapes were individual Zernike modes with a coefficient of 1 radian; the shapes applied are listed in *Appendix 6—table 1*. The RMS wavefront phase errors were calculated in the same way as before, and the results are shown in *Appendix 6—figure 1B*. They show that the device had already reached its steady state by the first data point at the 1 min mark. This indicated that the creep effect had a long-term component, as observed in the previous experiments, but there was also a significant short-term component which worked on the scale of seconds or even shorter. The nature of the creep effect depended on the current and the previously applied shapes, and as such, it acted as a memory effect where the final shape was a time-variant combination of the current and the previous shapes.

**Appendix 6—table 1.** Experimental conditions.

| Shape | Zernike mode (Noll index) | Holding time [min] |
|---|---|---|
| $S_0$ | $Z_4$ (defocus) | 240 |
| $S_1$ | $Z_5$ (primary oblique astigmatism) | 20 |
| $S_2$ | $Z_6$ (primary vertical astigmatism) | 20 |
| $S_3$ | $Z_7$ (primary vertical coma) | 20 |
| $S_4$ | $Z_8$ (primary horizontal coma) | 20 |
| $S_5$ | $Z_{11}$ (primary spherical) | 20 |
| $S_6$ | $Z_{22}$ (secondary spherical) | 60 |

To mitigate the creep effects, we adopted a set of exercise routines which helped with warming up of the device and with bringing its shape closer to its neutral stable position. We used a combination of three main exercises to diversify the way the reflective membrane was agitated. The first exercise consisted of alternating checkerboard patterns (*Appendix 6—figure 2A*), each held for 3 s, for a total of 50 repeats. The two inverted patterns typically covered 80% of the total actuator range, such that the two values were set at 10% and 90% of the full range. This exercise routine was especially helpful for removing any large errors in Zernike modes from the DM, which had arisen because large control voltages were applied to actuators for extended periods. The second exercise was cycling through varying levels of defocus (*Appendix 6—figure 2B*), with coefficients linearly distributed in the range $[-5; 5]$ rad, typically in 11 steps, each held for 100ms, for a total duration of about 10 min. This exercise helped to maintain the optimal PSF when using remote focusing. It could also restore

the optimal PSF if an extremely large amplitude of remote focusing was applied and affected the PSF quality. The third exercise (*Appendix 6—figure 2C*) consisted of the same type of actuator poking as in the calibration procedure. Each actuator was set individually to several values linearly distributed in the range $[10; 90]$% (80% of the total range). We typically used 11–21 steps, with one or two repeats. Because this exercise routine is identical to that used for Zernike mode calibration, it was best used for restoring optimal Zernike modes that are required for AO correction.

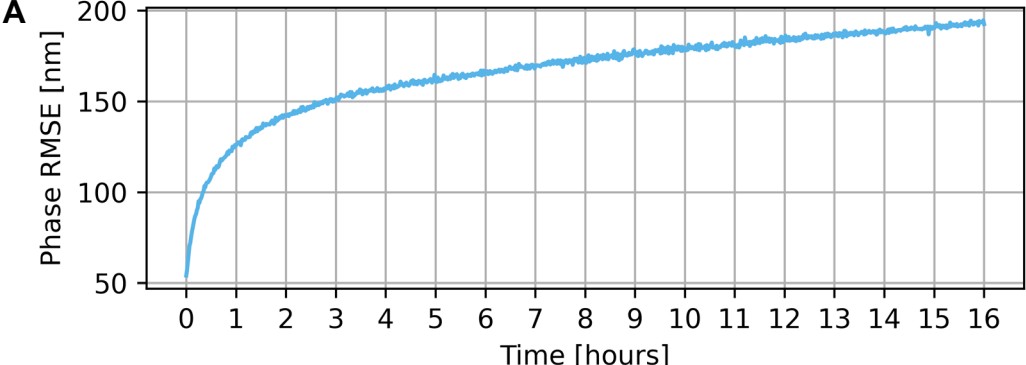

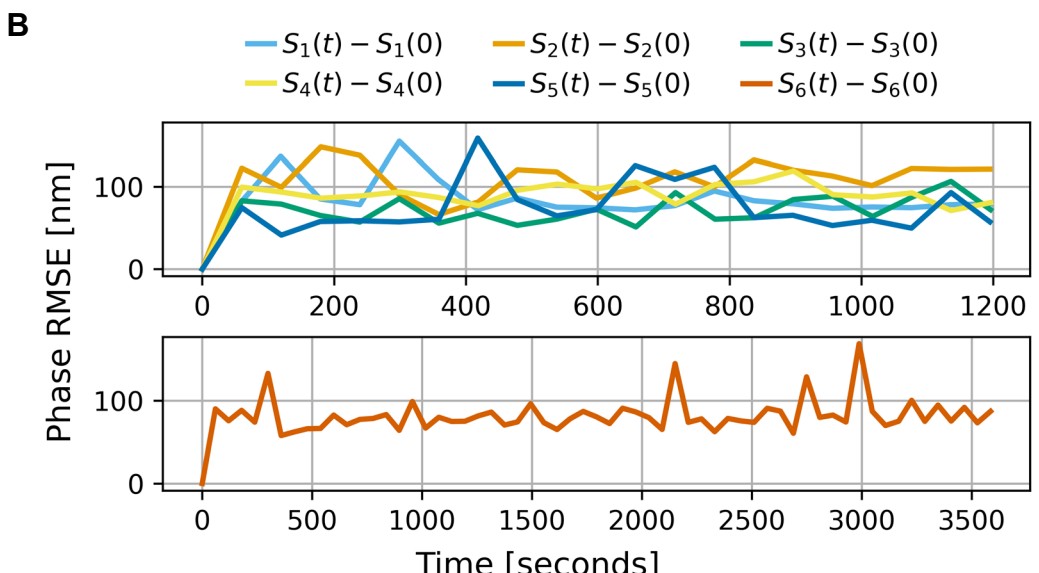

**Appendix 6—figure 1.** Deformable mirror (DM) drift and temperature characterisation. Wavefront phase root mean square (RMS) error (RMS error, RMSE) measurements for (**A**) long-term drift without warm-up and for (**B**) short-term drift with warm-up (4 hr of 1 rad defocus), following the sequences listed in *Appendix 6—table 1*. The two plots in (**B**) show shapes held for 20 min (top) and the final shape held for 60 min (bottom).

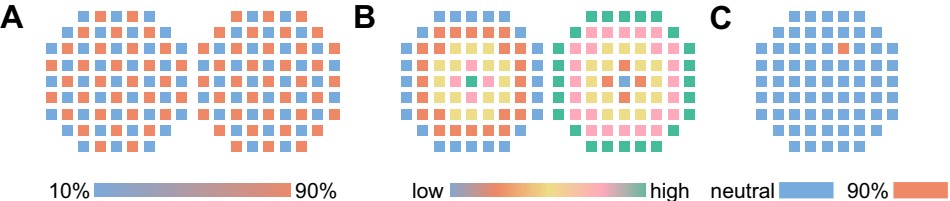

**Appendix 6—figure 2.** Actuator control signal patterns used in the deformable mirror (DM) exercise procedures. (**A**) Checkerboard. (**B**) Refocusing. (**C**) Individual poking.

