## [Editor Report]

Three-dimensional structured illumination microscopy (3D-SIM) is a technique that doubles imaging resolution. Still, its use has been limited by its sensitivity to aberrations in thick tissues and its lack of availability in an upright configuration. This convincing 'Deep3DSIM' method addresses these issues by using adaptive optics to correct aberrations and remote focusing for artefact-free volume imaging. This enables high-quality super-resolution imaging up to 130 µm deep into specimens, such as a *Drosophila* brain, while allowing for live-specimen manipulation, providing valuable advances on current efforts.

---

## [Decision Letter]

[Editors' note: this paper was reviewed by Review Commons.]

---

## [Author Response]

General statements

We thank the reviewers for spending so much time and care commenting on our manuscript. Their detailed, thoughtful and constructive comments have certainly improved the manuscript considerably. We have gone through each and every comment in detail and addressed it individually, often by new data and its analysis, leading to a revision of all the figures and supplementary figures of the paper. Below, we explain our responses and revisions in detail, outlining our response to each and every comment, closely following our revision plan, previously accepted by *eLife*. The revision of the figures now also follows more closely the guidelines of the journal. For example, single channel images are now presented in greyscale, while two-channel images use green and magenta. We also adhered to the principles of Colour Universal Design (by Okabe and Ito), whenever possible. In a similar vein, the “Supplementary Materials” have been correctly reorganised into appendices, now placed in the main manuscript document, and Figures 3-5 now include one supplementary figure each. The author contributions have been removed from the main manuscript, as well as the figure images, which are submitted as individual files. The figure legends remain in the main manuscript document.

Description of revisionsMajor reviewer commentsReviewer #1:Major comment #1

Given the emphasis on super-resolution imaging deep inside a sample, we were surprised to see no mention of other forms of structured illumination that allow super-resolution imaging in samples thicker than a single cell. These include the 'spot-scanning' implementations of SIM that offer better imaging at depth by virtue of pinholes, and include MSIM, iSIM, and rescan confocal technologies. The two-photon / AO implementation of iSIM seems particularly germane, e.g. https://pubmed.ncbi.nlm.nih.gov/28628128/ Please consider citing these works, as they help place the existing work into context.

We want to thank Reviewer #1 for making this good point. To address this comment, we added a few sentences in the introduction to describe and cite image scanning methods, referring to the work mentioned by the reviewer as well as additional studies:

“Image scanning methods have been used to enhance the spatial resolution in imaging of thick samples, in widefield (York et al., 2013; York et al., 2012), multiphoton (Ingaramo et al., 2014) and with AO (Zheng et al., 2017). These methods use physical or digital pinholes together with photon reassignment and deconvolution to improve the resolution by a factor of √2 or greater.”

Major comment #2As we're sure the authors appreciate, besides aberrations, a major additional obstacle to 3D SIM in thick tissues is the presence of out-of-focus background. Indeed, this point was mentioned by Gustafsson in his classic 2008 paper on 3D SIM (https://pubmed.ncbi.nlm.nih.gov/18326650/):'The application area of three-dimensional structured illumination microscopy overlaps with that of confocal microscopy, but the two techniques have different and complementary strengths. Structured illumination microscopy offers higher effective lateral resolution, because it concentrates much of the excitation light at the very highest illumination angles, which are most effective for encoding high-resolution information into the observed data, whereas confocal microscopy spreads out its illumination light more or-less uniformly over all available angles to form a focused beam. For very thick and compactly fluorescent samples, however, confocal microscopy has an advantage in that its pinhole removes out-of focus light physically. Structured illumination microscopy is quite effective at removing out-of-focus light computationally, because it is not subject to the missing-cone problem, but computational removal leaves behind the associated shot noise. Therefore confocal microscopy may be preferable on very thick and dense samples, for which the in-focus information in a conventional microscope image would be overwhelmed by out-of-focus light, whereas structured illumination microscopy may be superior in a regime of thinner or sparser samples.'This point is not mentioned at all in the manuscript, yet we are certain it is at least partially responsible for the residual image artifacts the authors mention. Please discuss the problem of out of focus light on 3D samples, particularly with an eye to the 'spot-scanning' papers mentioned above.

We thank Reviewer #1 for emphasising the importance of out-of-focus light in thick specimens and the difficulties it poses to good 3D-SIM imaging. To address the comment, we now describe this issue in the introduction, in the context of both 3D-SIM and image scanning methods:

“3D-SIM methods achieve higher resolution, but they manage background light less efficiently than image scanning methods. Two major reasons for this are the retention, after reconstruction, of the shot noise of the background light and the reduced contrast of the structured illumination (SI), potentially leading to reconstruction artefacts. These problems are compounded by depth and fluorescence density.”

Major comment #3The authors use a water dipping lens, yet they image into samples that are mounted on coverslips, i.e. they use a dipping lens to image through a coverslip:This almost certainly introduces spherical aberration, which the authors seem to observe: see attached pdf for reference.We find this troubling, as it seems that in the process of building their setup, the authors have made a choice of objective lens that introduces aberrations – that they later correct. At the very least, this point needs to be acknowledged in the manuscript (or please correct us if we're wrong) – as it renders the data in Figures 3-4 somewhat less compelling than if the authors used an objective lens that allowed correction through a coverglass, e.g. a water dipping lens with a correction collar. In other words, in the process of building their AO setup, the authors have introduced system aberrations that render the comparison with 3D SIM somewhat unfair. Ideally the authors would show a comparison with an objective lens that can image through a glass coverslip.

We thank Reviewer #1 for pointing out the confusion caused by our lack of clarity regarding the use of a water-dipping/immersion objective lens in our system. To address this comment, we have added descriptions in the introduction and the methods and materials sections that explain how the objective lens can be used in both water-immersion and water-dipping configurations:

“The system is based around a 60×/1.1 NA water-immersion objective lens, with a correction collar that allows its use in a water-dipping configuration without a cover slip.

We used a water-immersion objective lens (Olympus LUMFLN60XW), capable of working in a water-dipping configuration by adjusting a correction collar […]”

Major comment #4The authors tend to include numbers for resolution without statistics. This renders the comparisons meaningless in my opinion; ideally every number would have a mean and error bar associated with it. We have included specific examples in the minor comments below.

This is a good point, which we address below, in minor comments #8, #9, and #11. In summary, we revised Figure 2 to show the distributions of the FWHM samples, including common statistical measures such as median and mean, and other features like density, dispersion, and skewness.

Major comment #5In Figure 5, after the 'multipoint AO SIM', the SNR in some regions seems to decrease after AO: see attached pdf for referencePlease comment on this issue.

We want to thank Reviewer #1 for the insightful comment. We redesigned Figure 5, making sure that the two sets of images have the same intensity ranges. By doing this, the signal is now consistently higher in the AO images, so the issue of decreasing SNR is no longer present.

Major comment #6Please provide timing costs for the indirect AO methods used in the paper, so the reader understands how this time compares to the time required for taking a 3D SIM stack. In a similar vein, the authors in Lines 213-215, mention a 'disproportionate measurement time' when referring to the time required for AO correction at each plane – providing numbers here would be very useful to a reader, so they can judge for themselves what this means. What is the measurement time, why is it so long, and how does it compare to the time for 3D SIM? It would also be useful to provide a comparison between the time needed for AO correction at each (or two) planes without remote focusing (RF) vs. with RF, so the reader understands the relative temporal contributions of each part of the method. We would suggest, for the data shown in Figure 5, to report a) the time to acquire the whole stack without AO (3D SIM only); b) the time to acquire the data as shown; c) the time to acquire the AO stack without RF. This would help bolster the case for remote focusing in general; as is we are not sure we buy that this is a capability worth having, at least for the data shown in this paper.

We agree that the timing (and other) costs can be an important consideration, and we want to thank Reviewer #1 for bringing up this good point. To address this point, we have expanded the description of the AO in the methods and materials section to include more information about the time it takes to perform the different parts of the AO:

“The duration of the calibration routine was around 30 to 60 minutes, mostly spent on phase unwrapping and other intensive computations.

The duration of the correction routine for a single plane was proportional to the number of modes involved, the number of scanning points for each mode, and the camera exposure configuration, but usually in the order of seconds. The earlier example of 40 images for our base set of corrections would normally take around 10 seconds (<3.5 second of light exposure).

“

We also added a discussion of how the remote focusing could speed up the image acquisition, further elaborated in Appendix 3:

“In terms of speed, refocusing with the DM instead of using the piezo stage reduced the duration of the image acquisition significantly, because the settling time of the surface of the DM was around an order of magnitude faster than the settling time of the piezo stage, which highlights the potential of AO-based refocusing for fast imaging. A more detailed discussion of this point, together with a comparison between the DM and the piezo stage, is presented in Appendix 3.”

Major comment #7Some further discussion on possibly extending the remote focusing range would be helpful. We gather that limitations arose from an older model of the DM being used, due to creep effects. We also gather from the SI that edge effects at the periphery of the DM was also problematic. Are these limitations likely non-issues with modern DMs, and how much range could one reasonably expect to achieve as a result? We are wondering if the 10 μm range is a fundamental practical limitation or if in principle it could be extended with commercial DMs.

We are grateful to Reviewer #1 for the suggestions. During the revision of the manuscript, we noticed that there was an error in the reported range of the remote focusing in Figure 5. The actual range is ~16 µm, instead of 10 µm. In addition to this correction, we added discussion of the remote focusing range, the limitations of our approach, and whether other commercial DMs could improve the range (with more technical detail given in Appendix 2):

“The range of remote focusing shown in this work was limited to about ±8 µm, which is smaller than some previous demonstrations. There are two reasons why we did not use wider ranges in our results. First, the DM can only reliably produce PSFs with minimal aberration repeatedly up to a range of about ±10 µm (see Appendix 2). Second, for 3D-SIM acquisition, ±5 µm is already a wide and commonly used Z range. Especially in multi-colour imaging, a rather large number of images, about 1200 frames (3 angles × 5 phases × 80 Z steps) per channel, would be required. While larger volume acquisition is possible, it would lead to considerable photobleaching and phototoxicity, as well as longer acquisition and reconstruction times. Nevertheless, a similar DM device has been used to achieve an impressive range of remote focusing (Cui et al., 2021). Although their approach is different from the high-NA super-resolution imaging presented here, a stable DM should increase the practical range of our remote focusing approach twofold or even greater.”

Reviewer #2:Major comment #1The authors have provided an incomplete description of the structured illumination microscopy (SIM) reconstruction process. It is unclear whether the approach is based on 2D interference SIM configurations or 3D interference patterns. Furthermore, the specific algorithm utilized for image reconstruction has not been elucidated. Elaborating on these aspects is crucial as they significantly influence the interpretation of the resulting data.

We thank Reviewer #2 for highlighting the need to describe the reconstruction process in more detail. To address this point, we added further information about the reconstruction process in the Discussion section:

“We chose to use a well-established reconstruction method based on Wiener filtering, as implemented in the softWoRx software suite, without advanced pre-processing of the image data for reduction of artefacts.”

And in the methods and materials section:

“For SIM data reconstruction, we used the commercial software softWoRx (GE Healthcare), which uses Wiener deconvolution, based on the approach by (Gustafsson et al., 2008).”

Major comment #2The authors have stated that sample-induced aberrations caused by RI inhomogeneities within the specimen is another major reason for causing artifacts generation. Literature has demonstrated that RI inhomogeneities can lead to non-local distortions in the grid pattern, which suggests that applying uniform reconstruction parameters across the entire image may not be viable. Traditional artifact remediation using the classical Wiener method is likely insufficient under these conditions (PMID: 33896197). The existing adaptive optics (AO) approach, which employs a deformable mirror (DM) alongside an sCMOS camera, is inadequate for tackling the issue at hand. Actually the assertion made in the paper that "aberrations change approximately linearly with depth" is seemingly contradicted by simulations referenced in the cited literature (PMID: 33896197). Consequently, it appears that the current methodology might only achieve a partial mitigation of the problems associated with spherical aberration resulting from RI mismatches. It is advisable, therefore, that the authors explicitly acknowledge this limitation in their manuscript to prevent any potential misinterpretation by readers.

We are thankful for this thoughtful comment by Reviewer #2. The focus of our work was not the use of advanced 3D-SIM reconstruction and aberration correction methods; instead, we used standard ones that do not deal perfectly with field-dependent aberrations. As such, our approach provides average reconstruction and average correction across the entire field of view. To address the comment, we have added a new paragraph to the Discussion section:

“Our AO methods provide only one average correction […]”

We also highlighted specific examples of residual aberrations in Figure 4B.

Major comment #3In Figure 2, the use of COS-7 cells, which are known for their relatively thin axial dimension, for the experiments raises an eyebrow. Notably, there are ample instances in existing research where both 2D-SIM and 3D-SIM, without the integration of adaptive optics, have yielded high-quality super-resolution images of structures such as tubulin and the endoplasmic reticulum. In addition, the authors did not present a direct comparison between BP-SIM and AO-SIM here. Without this comparative analysis, it remains ambiguous whether the enhancements in resolution and contrast and the reduction in artifacts can genuinely be attributed to the mitigation of spherical aberration. To clarify this, it would be beneficial for the authors to include side-by-side comparisons of these modalities to demonstrate the specific improvements attributed to AO-SIM.

We are grateful to Reviewer #2 for this helpful comment. We revised Figure 2 entirely, which includes the bottom part of it (Figure 2B), to emphasise better the comparison that we wanted to make: widefield against 3D-SIM, demonstrating the improvement in resolution. Naturally, we also revised the second paragraph of the Results section accordingly, to be in line with the updated figure and to make our arguments clearer.

Major comment #4In Figures 3 and 4, the authors have illustrated the enhancements achieved through the application of AO. However, there is a discernible presence of hammer-stroke and honeycomb artifacts in pre-AO imaged data, which seem to originate from the amplification of the incorrectly moved out-of-focal background in the frequency domain. Various strategies have been previously suggested to address these specific artifacts, encompassing methods like subtracting background noise in the raw images or employing selective frequency spectrum attenuation techniques, such as Notch filtering and High-Fidelity SIM. To facilitate a more comprehensive understanding, I would recommend that the authors incorporate into their study a comparison that includes BP-SIM data that has undergone either background subtraction or frequency spectrum attenuation. This added data would enable a more complete evaluation and comparison regarding the merits and impact of their AO approach.

We thank the reviewer for this excellent suggestion. We agree that a pre-processing step, such as background subtraction or frequency spectrum attenuation, can help with the reduction of artefacts. However, when we tried frequency spectrum attenuation, as we said in the revision plan, the results did not improve the image quality. We note that the methods suggested by the reviewer, e.g. HiFi-SIM, were designed for 2D-SIM and they are not suitable for our 3D-SIM methodology. We feel that the effective removal of 3D-SIM reconstruction artefacts is a complex and difficult task, which deserves its own treatment and therefore it is well beyond the scope of this manuscript. Nevertheless, to address this comment, we have added a new paragraph to the Discussion section, which clarifies our reconstruction strategy and choice of algorithms:

“An important consideration in 3D-SIM reconstruction is the precise settings […]”

The paragraph also includes references to novel 3D-SIM reconstruction methods which could produce higher fidelity reconstructed data, with less artefacts, and which can be applied to Deep3DSIM.

Reviewer #3:Major comment #1There is an overall reference in the manuscript of the novelty possible range of applications of using an upright microscope configuration. Examples mentioned are tissue-based imaging, access to whole-mount specimens for manipulation and electrophysiology. However, authors fail to present any such applications. There is not a single example presented which could not have been obtained with an inverted microscope. Could the authors provide an example where a water-dipping is used. Expanded samples could be one case, since the thickness of the gel makes it difficult to image with an inverted microscope. Another possible example would be to label the extracellular space and do shadow imaging of the tissue (SUSHI PMID: 29474910). ExM might be simpler to do as part of revising the manuscript than SUSHI.

We are thankful to Reviewer #3 for the constructive comment. The data in Figure 5A is an example where water-dipping is used, which we now explicitly state in the revised figure legend. The reviewer’s notes on expansion microscopy and SUSHI are interesting, but the primary purpose of our microscope system is to facilitate live super-resolution cell imaging experiments. Nevertheless, to address this comment, we have added a discussion of the application of Deep3DSIM to expansion microscopy:

“Deep3DSIM is also uniquely positioned for the imaging of the whole sample in expansion microscopy, without the need for mechanical sectioning, which is a challenge because of the softness and fragility of commonly used hydrogel techniques (Cahoon et al., 2017).”

Major comment #2On the main text it is described a 5-fold volumetric resolution, which is confusing since authors only mention lateral and axial resolutions. Their measurements correspond to a ~1.6-fold lateral improvement and ~1.7-fold axial improvement. These are however not the 95% of the achievable resolution theoretical maximum, as stated in p7 SI (2 fold increase of 282nm), but only the 80-85%. This point should be rephrased in the manuscript.

We thank Reviewer #3 for bringing up this important point. In response, we have revised our description of the resolution improvement in the first two paragraphs of the Results section. We also removed the claim of 5-fold improvement, which we agree is confusing.

Major comment #3[OPTIONAL] p4 and related to figure 2, it would be important to report also measurements of beads with SIM but without AO, just as done for WF. Is there an improvement of using AO on SIM? This is reported for the fixed cells but not for the beads.

We agree with the reviewer’s point and in response we have revised Figure 2, for greater clarity. We wanted to demonstrate the resolution improvement of 3D-SIM over widefield imaging. With the revision of the figure, we shifted the emphasis away from the AO to avoid confusion.

Major comment #4Figure 2, it is odd the comparison between WF+/- AO and SIM +/- AO are done using different cellular structures. Since wavelengths used are not the same it is difficult to interpret if there is any improvement of using AO on SIM compared to SIM without AO. Same questions arise as above, Is there an improvement of using AO on SIM?

We agree that the data in Figure 2C-D is presented in unusual way. We addressed the comment by revising Figure 2 to make it clear from this point of view. We also changed considerably the corresponding descriptions in the first two paragraphs of the Results section.

Major comment #5"A significant benefit and uniqueness of the Deep3DSIM design is its upright configuration, whereas commercial SIM systems are built around inverted microscopes and are usually restricted to imaging thin samples, such as cultured cells." (p5) is not correct. The commercial DeepSIM module from CREST Optics can be mounted on an inverted microscope as well as image deep into tissue (seehttps://crestoptics.com/deepsim/ and application notes therein) and be used with essentially any objective. This point should be rephrased in the text.

We thank Reviewer #3 for pointing out this mistake. Of course, we meant commercial 3D-SIM systems, such as GE Healthcare DeltaVision OMX and Nikon N-SIM. We have rephrased this sentence in the beginning of the third paragraph of the Results section:

“Deep3DSIM is an upright system with a long working distance objective lens, in contrast to commercial 3D-SIM systems which are built around inverted microscopes, and which are usually restricted to imaging thin samples, such as cultured cells.”

Regarding the commercial DeepSIM module from CREST Optics, as far as we can tell, it uses an imaging scanning microscopy method. This is very different from our method, which uses 3D-SIM.

Major comment #6Figure 3 reports the improvements of AO on SIM for imaging over 10um in tissue. What are the Zernike modes measured? Or how does the pupil look like before and after correction? It would be also good to report the Fourier amplitudes as done in Figure 2C as a quantitative measure of improvement. It would be good to point out the artifacts observed on the BP SIM image reconstruction (labelled with 3x, fringes are noticeable).

This is a good suggestion – thanks. In response, we have revised Figure 3 to improve the presentation of the data, which includes the pointing out of reconstruction artefacts. We also added Figure 3—figure supplement 1 to show the power spectra and the Zernike mode corrections.

Major comment #7Many key details relating to image acquisition and AO correction are missing for all figures. How is the AO optimization implemented? Is it implemented via a genetic algorithm (progressive optimization of parameters) or using more clever strategies? Not clear if the optimization is implemented using images obtained with flat illumination or after SIM imaging/processing of a given dataset. How long does the AO optimization take? How sensitive to noise is the process? What metric do they use to estimate the sensorless AO correction? On pag12, they say "Fourier domain image metric" for measurements with fine details; otherwise, ISOsense when not high frequencies are present. Could the authors report the formula used to calculate the first metric? What do they consider to be low and high frequencies in this case? Is there a reason why ISOsense is not always used, or is there an automatic way to choose between the two? How many images were acquired for AO correction? Which samples were corrected with ISOsense and which ones with Fourier domain image metric? (see for example the detailed experimental reporting in the Supp Mat from Lin et al. Nat Commun 2021).

We are grateful to Reviewer #3 for the extensive list of questions. We greatly expanded our description of the AO methods in the methods and materials section, which now answers all these questions. We further added a new appendix (Appendix 5) where we provide formulae and description of the Fourier-based image metric.

Major comment #8Figure 4. Data presented for larval brain tissue is a very clear example of adding AO to image deep into tissue as the effect at ~130 cannot be understated. Here too, it would be also good to report the Fourier amplitudes as done in Figure 2C as a quantitative measure of improvement and possibly the SNR of reconstructed images. Having a way to quantitatively describe how much better are those images would be great. Also, what are the aberrations corrected? Can the wavefront or Zernike amplitude of the modes be reported? Same as for Figure 3, details about AO correction are missing.

This is a very helpful comment – thanks. We added Figure 4—figure supplement 1, which shows the power spectra and the Zernike correction modes.

Major comment #9

[OPTIONAL] "It is worth noting that aberrations can differ across larger fields, and therefore, after applying an average correction, residual aberrations can still be observed in some regions of the AO-corrected bead images. However, the overall PSF shapes were still dramatically improved with AO compared to the equivalent without AO." This point is very interesting although not result either in the main text or in the SI is presented.

This is a good point. In response, we revised Figure 4, to highlight two examples of residual aberrations. We also added a new paragraph to the Discussion section, where we elaborate on the subject:

“Our AO methods provide only one average correction […]”

Major comment #10"As we found that the aberrations change approximately linearly in depth, we could measure the aberration in two planes and calculate the corrections in intermediate planes by interpolation, an approach which we termed "multi-position AO"." This is, personally, one of the major contributions of this work to the community. Unfortunately, it is not reported in detail. Not only for SIM but for imaging with WF or confocal, such linear change for aberrations with depth is not well known. Again, here the details of AO correction and image metrics are missing. To establish that for most thick biological structures 'aberrations change approximately linearly in depth' would be foundational to the widespread use of AO within standard imaging. Would it be possible for the authors to elaborate on this point and present detailed results? What is the error from measuring and correcting more than 2 planes? What is the error from just measuring and AO correcting at the deeper plane, i.e. from a single measurement? Authors could also show a case in which a linear assumption works nicely (or how well it works). For example, comparing an intermediate plane (or a plane beyond) imaged after AO optimization or after coefficient interpolation of the Zernike modes and compare it against correcting directly that plane.

We thank the reviewer for highlighting our poorly phrased sentence. We meant to say proportional instead of linear. We have rephrased our argument, which is now in the last paragraph of the Results section. In general, our description of the AO methods has been greatly expanded and includes details about the AO correction and the image metrics. Performing estimation from just a single measurement is not possible, but we show the simplest possible example – line fitted to two measurements (planes) – in Figure 5B, which also shows planes beyond the deep measurement point (bottom arrow).

Major comment #11The image of the cos-7 cell in metaphase, for Figure 5 is, however, very disappointing. See Figure 1 of Novak et al. Nat Commun 2018 for an example of a single z-plane of a cell in metaphase. Having the possibility to correct for the entire 3D volume, I would expect amazing 3D volumes (movies and/or projections) associated with this imaging which are not presented.

This is an interesting point. In response, we revised the old Figure 5 to more effectively display the data, including clearer line profile measurements. The revised figure was merged with the old Figure 6, and it now corresponds to Figure 5B. This revision now allows us to motivate and describe the experiments better in the last two paragraphs of the Results section.

Major comment #12In Figure 6, they use AO in remote configuration mode to allow imaging of live specimens. It needs to be clarified if this is an a priori characterization that is then kept fixed while recording in time. The last acquired volume of Figure 6A and B have a higher amount of artifacts with respect to time 00:00. Are those artifacts due to lower SNR (maybe due to sample bleaching) or due to some change in the aberrations of the specimen?

Thanks for this valuable comment. Our revised description of the AO methodology now explicitly states that the AO correction characterisation was performed before the actual data acquisition:

“When imaging, we always carried out the aberration correction routine first and then held the DM in the corrective shape during the actual image acquisition. For relatively thin volumes, e.g. 1-2 µm thickness, we mostly corrected just a single plane (usually the middle of the stack) […]”

Regarding the last volume in Figure 6 (now Figure 5A), the reorganisation of the mitotic spindle are highly dynamic, but the reconstruction artefacts remain constant throughout the series. Nevertheless, for clarity and transparency, we pointed the artefacts out in the description of the results, and we refer the readers to the methods and materials section, where we added a new subsection on this topic, called “SIM reconstruction artefacts”.

Major comment #13"These results demonstrate that the remote focusing functionality of the system can be successfully applied for live 3D-SIM experiments, allowing four-dimensional acquisition while keeping the specimen stationary, thus avoiding the usual agitation and perturbations associated with mechanical actuation." Generally, this statement is true, but for the specific example shown of Drosophila embryogenesis is it relevant? If they use piezo-driven Z-stack imaging with AO, does that lead to incorrect reconstructions or motion-induced artifacts? Related to the results shown in Figure 6, the fair comparison would be AO SIM vs SIM (without AO), not AO SIM vs AO WF.

We have addressed this excellent comment by revising Figure 6 (now Figure 5A) and its description in the penultimate paragraph on the Results section. We hope this revision clarifies that remote focusing can meet the stringent image quality requirements of 3D-SIM reconstruction and that it is suitable for live imaging, i.e. fast and gentle with the excitation light.

Major comment #14When performing remote focusing, is the effective NA of the imaged plane changing with respect to the NA of the objective used at its focal plane?

To address this good point, we now mention the negligible change in effective NA in the methods and materials section:

“The change in effective NA due to focus shift was negligible with our choice of focusing range (up to ~16 µm) and objective lens.”

Major comment #15[OPTIONAL] Did the authors run calculations to explore whether a commercial upright microscope could be used instead of their design? Are there any fundamental flaws that would make impossible using a commercial base? If not, could an AO SIM module be designed such that it adds on a commercial base? It would be important to discuss this point.

We thank Reviewer #3 for bringing up this interesting point. A lot of considerations, calculations, and modelling were done in the prototyping of Deep3DSIM. Of course, the use of a commercial upright microscope stand was considered. One of the obvious limitations was the difficult access to the pupil-conjugated plane. In addition, a commercial microscope stand was not compatible with many of the core parts of the system, which were chosen for specific biological applications, such as dual camera system for fast live simultaneous imaging and the heavy-duty Z stage intended to support two heavy micromanipulators. To address this point, we added a brief discussion of the commercialisation of Deep3DSIM to the last paragraph of the Discussion section:

“Finally, Deep3DSIM was custom prototyped on a large optical table, instead of basing the design around a commercial upright microscope, because the latter would have imposed unnecessary restrictions at this prototyping stage. In the future, the optical path of Deep3DSIM could be simplified and reduced in size, thus making it easier to adopt or commercialise, and to add on to existing microscopes, as well as to build specific Deep3DSIM systems designed for bespoke applications.”

Minor reviewer commentsReviewer #1:Minor comment #1The paper mentions Ephys multiple times, even putting micromanipulators into Figure 1 – although it is not actually used in this paper. If including in Figure 1, please make it clear that these additional components are aspirational and not actually used in the paper.

To address this point, we revised the legend of Figure 1 to make it clear that the use of the micromanipulators is conceptual and not actually shown in the manuscript.

Minor comment #2The abstract mentions '3D SIM microscopes', 'microscopes' redundant as the 'm' in 'SIM' stands for 'microscope'.

To correct this error, we have revised the wording in the abstract and other parts of the manuscript that mentioned “SIM microscope(s)”, which we replaced with “SIM system(s)”.

Minor comment #3'fast optical sectioning', line 42, how can optical sectioning be 'fast'? Do they mean rapid imaging with optical sectinong?

Yes. We have now revised this part of the introduction to “fast imaging with optical sectioning”.

Minor comment #4line 59, 'effective imaging depth may be increased to some extent using silicone immersion objectives', what about water immersion objectives? We would guess these could also be used.

We rephrased this part of the introduction to include water immersion objective lenses:

“Silicone oil and water immersion, or water-dipping, objective lenses, with lower NA, have longer working distances […]”

Minor comment #5line 65 – evidence for 'water-dipping objectives are more sensitive to aberrations' ? Please provide citation or remove. They are certainly more prone to aberrations if used with a coverslip as done here.

We reviewed this phrase and we decided that it is poorly phrased and confusing. Therefore, we removed it.

Minor comment #6'fast z stacks' is mentioned in line 103. How fast is fast?

We changed the wording of this line to make it clearer that we are referring to acquisition speed. We discuss the acquisition speed in the last paragraph of the Discussion section, and we provide more technical details about the axial scanning speed in Appendix 3.

Minor comment #7line 116 'we imaged 100 nm diameter green fluorescent beads'. Deposited on glass? Given that this paper is about imaging deep this detail seems worth specifying in the main text.

We added this detail, namely that the beads were deposited on a glass cover slip, to the first paragraph of the Results section.

Minor comment #8lines 127-130, when describing changes in the bead shape with numbers for the FWHM, please provide statistics – quoting single numbers for comparison is almost useless and we cannot conclude that there is a meaningful improvement without statistics.

We agree with this comment, so we have provided statistics for the numbers from the first paragraph of the Results section. In addition, we present thorough statistical treatment of the FWHM measurements in Figure 2A.

Minor comment #9In the same vein, how can we understand that remote focus actually improves the axial FWHM of the widefield bead? Is this result repeatable, or it just noise?

Yes, the remote focusing does improve the axial FWHM resolution, compared to the AO bypass case, since our implementation of the remote focusing corrects for some aberrations, such as spherical aberration. However, we removed the remote focusing FWHM resolution estimations from Figure 2A, because they detracted from the main comparison that we wanted to make in this figure – widefield vs SIM.

Minor comment #10line 155, 'Because of the high spatial information…' -> 'Because of the high resolution spatial information…'

We removed this sentence from revised manuscript.

Minor comment #11When quoting estimated resolution #s from microtubules (lines 158-163) similarly please provide statistics as for beads.

We further considered this suggestion and, on reflection, we realised that our estimate of resolution uses the frequency response of the imaging system, which does not lend itself to the same statistical treatment as the FWHM resolution estimations. To make this point clearer, we revised the second part of Figure 2 extensively, now showing explicitly how the numbers were derived. We also rephrased the respective description in the text (second paragraph of Results section) and the figure’s legend.

Minor comment #12It seems worth mentioning the mechanism of AO correction (i.e. indirect sensing) in the main body of the text, not just the methods.

We agree with this comment. We added a brief description of the AO approach to the third paragraph of the Results section:

“Our implementation of the AO did not use a dedicated wavefront sensor, instead relying on an indirect sensorless approach.”

Minor comment #13How long do the AO corrections take for the datasets in the paper?

We added details regarding the duration of the calibration and the AO-based corrections to the methods and materials section.

Minor comment #14Were the datasets in Figure 2-4 acquired with remote focusing, or in conventional z stack mode? Please clarify this point in the main text and the figure captions.

No, they were not. To address this comment, we rearranged the results, and we merged the old Figure 5 and Figure 6. The new Figure 5 is now the only example with remote focusing.

Minor comment #15It would be helpful when showing z projections in Figures 3-5 to indicate the direction of increasing depth (we assume this is 'down' due to the upright setup, but this would be good to clarify)

We revised all figures to have axis arrows for the microscope images, indicating the direction of increasing depth for images with Z axis.

Minor comment #16line 174, 'showed significant improvements in both intensity and contrast after reconstruction' – we see the improvements in contrast and resolution, it is harder to appreciate improvements in intensity. Perhaps if the authors showed some line profiles or otherwise quantified intensity this would be easier to appreciate.

We revised all figures to either have the same intensity ranges, making it easy to visually appreciate changes in intensity, or to have bars (usually above the image) that quantify the intensity range.

Minor comment #17line 195 'reduced artefacts' due to AO. We would agree with this statement – the benefit from AO is obvious, and yet there are still artefacts. If the authors could clarify what these (residual) artefacts are, and their cause (out of focus light, uncorrected residual aberrations, etc) this would be helpful for a reader that is not used to looking at 3D SIM images.

We agree with this comment. To address it, we explicitly pointed out the residual aberrations in the results of Figure 4B. We also consolidated the discussion of this topic to new subsection (“SIM reconstruction artefacts”), which we added to the Methods and materials section.

Minor comment #18Line 197, 'expected overall structure', please clarify what is expected about the structure and why.

We took out from the legend of Figure 4 our brief description of the expression pattern of the Cno protein, we greatly expanded it, and put it in the Results section:

“The larvae used to prepare the samples came from a fly line with YFP protein trap for canoe (cno), which encodes a protein that is expressed in the adherens junctions of the monolayer of neuroepithelial stem cells in the outer proliferation centres in the optic lobes. The protein expression marked the outlines of the cells and showed the characteristic geometrical mosaic pattern that emerges from the packing of epithelial cells, covering large parts of each brain lobe and thus providing a well-defined structure that could be imaged at various depths.”

We also changed the “expected overall structure” phrase to “expected overall mosaic structure”.

Minor comment #19Line 199, what is a 'pseudo structure'?

We replaced this poor description with “fuzzy patches”.

Minor comment #20Figure 4B, 'a resolution of ~200 nm is retained at depth', please clarify how this estimate was obtained, ideally with statistics.

We agree with this comment, and have removed the confusing statement. What we meant was that the lateral FWHM resolution at ~130 µm is mostly the same as on the surface of the sample, which we now explicitly show in Figure 4C. We now also state in the results that “the lateral FWHM resolution is preserved even at this depth”.

Minor comment #21Figure 4D, please comment on the unphysical negative valued intensities in Figure 4D, ideally explaining their presence in the caption. It would also be helpful to highlight where in the figure these plots arise, so the reader can visually follow along.

We agree with this comment. The negative intensities were removed in the revision of Figure 4 and therefore, an explanation is no longer necessary. We also indicated in the images where the line profiles came from.

Minor comment #22Line 245, 'rapid mitosis'. What does rapid mean, i.e. please provide the expected timescale for mitosis.

The mitotic cycles at this developmental stage are short, e.g. 5 minutes per mitosis, compared to those of somatic cells, where it takes several hours. We expanded our description of this developmental stage, specifying the duration of each division cycle:

“[…] we imaged *Drosophila* embryos undergoing rapid mitosis, where each division cycle lasts just a few minutes […]”

Minor comment #23For the data in Figure 6, was remote refocusing necessary?

We improved our motivation for the data in Figure 6 (now Figure 5A) in the penultimate paragraph of the Results section:

“A key feature of Deep3DSIM is the ability to perform remote focusing. On the other hand, the reconstruction process of 3D-SIM is very stringent about the image quality and the image acquisition parameters, such as the accuracy and precision of Z steps in the acquisition of Z-stacks. Therefore, we wanted to show that remote focusing can meet the strict requirements of 3D-SIM reconstruction, and that the remote focusing can be used as a direct replacement of a mechanical Z stage over a moderate range. We demonstrate this with two examples, starting first with live imaging. Living specimen already present several challenges for 3D-SIM imaging on their own (Gao et al., 2012). Particularly important are the acquisition speed, which can lead to motion artefacts if not sufficiently high, and photobleaching and phototoxicity, which occur when the rate of exposure and the intensity of the excitation light are too high. The Deep3DSIM approach has the potential to resolve both these challenges by using remote focusing for fast image acquisition while preserving the high image quality of the system.”

Minor comment #24What is the evidence for 'reduced residual aberrations', was a comparative stack taken without AO? In general we feel that the results shown in Figure 6 would be stronger if there were comparative results shown without AO (or remote focusing).

We agree with this comment, so we removed the confusing statement. However, we did add a new paragraph to the Discussion section, where we discuss field-dependent aberrations and the residual aberrations in Figure 4B:

“Our AO methods provide only one average correction […]”

Minor comment #25Line 350, 'incorporation of denoising algorithms' – citations would be helpful here.

We added two examples of such denoising algorithms to the last paragraph of the Discussion section:

“[…] future improvements could also include the incorporation of denoising algorithms into the reconstruction process to allow lower light dosage, which would enable faster imaging, reduced photobleaching, and reduced specimen photodamage, as demonstrated, for example, in (Huang et al., 2018; Smith et al., 2021).”

Minor comment #26Line 411, 'All three were further developed and improved' – vague, how so?

We agree with this comment. To address it, we added a brief description of the notable changes and we added references to software repositories where more detailed descriptions can be found:

“All three were further developed and improved, to match the requirements of Deep3DSIM. Most notably, remote focusing was added to the microscope-aotools package. All changes were tracked and they are described in more detail in the respective branch repositories:

python-microscope: https://github.com/dstoychev/microscope/tree/deepsim

microscope-cockpit: https://github.com/dstoychev/cockpit/tree/deepsim

microscope-aotools: https://github.com/dstoychev/microscope-aotools/tree/deepsim-matthew-danny.”

Minor comment #27Sensorless AO description; how many Zernike modes were corrected?

We specified which Zernike modes were usually corrected in the methods and materials section. Instead of tables, we have now added proper plots of example correction modes to the figure supplements of Figure 3 to Figure 5.

Minor comment #28Multi-position aberration correction. Was the assumption of linearity in the Zernike correction verified or met? Why is this a reasonable assumption?

The claim of linearity arose from poor phrasing. What we meant was proportionality with depth and linearity of the estimation. We have now rephrased that section in the last paragraph of the Results section:

“We demonstrate MPAC in its simplest form, a linear estimation of two planes, which we found to work well, partially because some of the dominant aberrations, like spherical, are proportional to the imaging depth.”

Minor comment #29Figure S1B is not useful; if the idea is to give a visual impression of the setup, we would recommend providing more photos with approximate distances indicated so that the reader has a sense of the scale of the setup. As is – it looks like a photograph of some generic optical setup.

Figure S1 is now Appendix 4 Figure 1. Despite our intention to provide more photographs of the instrument, we discovered that its footprint is too large and complex to adequately capture in a single photograph, especially within the dimensions that the figure would allow. We decided against the addition of multiple photographs and therefore removed the second part of the figure entirely, to avoid confusing the readers with photographs that are difficult to understand.

Minor comment #30SI pattern generation – 'the maximum achievable reconstruction resolution was only slightly reduced to about 95% of the theoretical maximum'. We don't understand this sentence, as the resolution obtained on the 100 nm beads is considerably worse than 95% of the theoretical maximum. Or do the authors mean 95% of the theoretical maximum given their pitch size of 317 nm for green and 367 nm for red?

We agree with this comment, so we have removed the confusing phrase.

Minor comment #31SI Deformable mirror calibration 'spanning the range [0.1, 0.9]' – what are the units here?

This comment is related to part of what used to be the supplementary material, describing one of the calibration procedures of the AO. To avoid confusion, we decided to remove the details about calibration procedure, as they were already well documented in the cited literature, e.g. (Antonello et al., 2020; Hall, 2020).

Minor comment #32What are the units in Figure S5C, S5D?

Similar to minor comment #31, this figure was removed to avoid possible confusions.

Minor comment #33It would be useful to define 'warmup' also in the caption of SI Figure S6A.

We agree with this comment. We added a description of the warm-up routine to the figure legend: “warm-up (4 hours of 1 rad defocus)”.

Minor comment #34SI Remote Focusing, 'four offsets, {-5 mm, -2.5 mm, 2.5 mm, 5 mm}…' are the units mm or um?

The units were supposed to be micrometres. We corrected this error: “four offsets, {-5, -2.5, 2.5, 5} µm”.

Minor comment #35'…whereas that of the 10 beads was…' here, do the authors mean the position of the beads derived from the movement of the piezo stage, as opposed to the remote focusing?We agree with this comment. To address this confusion, we have re-analysed the data using exact pooled variance for each of the four offset steps, providing a standard well-defined measure of dispersion:

“In terms of precision, we calculated the exact pooled variance of each of the offsets to be {6.45, 4.15, 20.58, 11.83} nm, respectively.”

Minor comment #36The authors refer to the 'results from Chapter 3.2'. What are they talking about? Do they mean a supplementary figure, or earlier supplementary results? In general, we found the discussion in this paragraph difficult to follow.

This was a remnant from an earlier version of the supplementary material, which used numbered sectioning. We rephrased this part from “results from Chapter 3.2” to “findings from Appendix 6”.

Minor comment #37Supplementary Figure 9 seems to be not referred to anywhere in the text.

To correct this, we changed Supplementary Figure S9 to be Figure 5—figure supplement 1, and we refer to in the methods and materials section:

“Example correction modes and their amplitudes for each sample are given in the figure supplements of Figure 3 to Figure 5.”

Minor comment #38Since the paper emphasizes 3D SIM, OTFs along the axial direction would also be useful to show, in addition to the lateral OTFs shown in Figure 2D.

To address this point, we have added axial power spectra, showing the axial OTF supports, to Figure 2. We also provide similar data in the figure supplements for Figure 3 to Figure 5.

Minor comment #39When the sample is moved by the piezo, the axial phase of the 3D-SIM illumination pattern is stable as the sample is scanned through the illumination pattern. When remote focusing is performed, the sample is always stable so the axial phase of the 3D-SIM illumination pattern is presumably changing with remote focusing. Can the authors clarify if the 3D SIM illumination pattern is scanned when remote focusing is applied, or is the intensity pattern stable in z?

To address this point, we have now clarified in the methods and materials section that the structured illumination pattern was scanned:

“When remote focusing was combined with 3D-SIM imaging, the focal plane and the SI pattern were synchronously moved together during axial scanning, keeping constant the phase relationship between the two.”

Minor comment #40

In Supplementary Figure 9, primary spherical is referred to twice, both at index 11 and 22. The latter is presumably secondary spherical?

Yes. We revised Supplementary Figure S9, which is now Figure 5—figure supplement 1, and we removed the removed the arrows to avoid confusion.

Minor comment #41We do not understand the x axis label, in Figure S4D, is it really [0, 50, 50, 50] as written?

We thank the review for pointing this out, as the x axis was mislabelled. It is supposed to show three 0 to 50 μm ranges. Supplementary Figure S4 is now Appendix 1—figure 1, and we modified the x-axis labels in panel B to correctly show the full range of 150 µm. We also expanded our description of the plot in the figure legend:

“The plot shows three 50 μm scans through the bead, one for each of the SI angles.”

Reviewer #2:NoneReviewer #3:Minor comment #1Figure 2 lacks a color bar for D panels, which is in log scale. Authors should also show the Fourier transform along the z direction.

To avoid confusion, the colour in the power spectra of Figure 2 has been removed, so the need for a colour bar is no longer necessary. However, we did specify in the figure legend that the images are in logarithmic scale. We also added the axial power spectra.

Minor comment #2p4, "Such minor aberrations tend to be insignificant in conventional microscopy modalities such as widefield and confocal (Wang and Zhang, 2021). Therefore…" If optical aberrations are insignificant for single cells in widefield and confocal why do experiments here? These sentences should be rephrased to motivate better the experiments performed.

We agree with this comment, and we rephrased the sentence:

“Such minor aberrations are often overlooked in conventional microscopy modalities such as widefield and confocal (Wang & Zhang, 2021). Therefore […]”

Minor comment #3Imaged microtubules look abnormal, 'dotty' (figure 2) in both WF and SIM. See https://sim.hms.harvard.edu/portfolio/microtubules/ or Figure 1 of Wegel, et al. Dobbie Sci Rep 2016, for better examples of continuous microtubule structures as imaged with SIM.

To address this point, we have added a new paragraph to the “Cell culture” part of the methods and materials section, where we address the fragmentation of microtubules:

“The microtubules stained in this way, occasionally appeared fragmented, more noticeable in the SIM images in Figure 2B. This fragmentation is normal for a sample preparation protocol, such as ours, based on 2% paraformaldehyde fixation. More advanced protocols, specifically designed for the preservation of structures such as microtubules, can result in more continuous filaments.”

Minor comment #4Is also the remote focusing performed via optimization of metrics similar to the one used for compensating aberrations?

Yes, the remote focusing is based on our regular aberration correction method. To improve clarity, we rephrased our description of the remote focusing in the methods and materials section, explicitly stating that its calibration uses the same type of optimisation algorithm as the aberration correction:

“To calculate the required remote focusing DM patterns, we first performed a calibration procedure, which was based on our aberration correction method and hence used the same type of optimisation algorithm.”

We further elaborate on this procedure in Appendix 2.

Minor comment #5Figure 2, the order of names on the top right of the panel should match the order of curves presented.

The plot legends were removed during the revision of Figure 2, thus making this comment no longer applicable to the new figure.

Minor comment #6I value the efforts to improve open-source tools for system and AO control and GUI. And those tools seemed to have been modified for this work, although those modifications are not described. Would it be possible for the authors to describe those modifications?

We added a brief description of the major changes, as well as references to the software repositories where all those changes were tracked and documented:

“All three were further developed and improved, to match the requirements of Deep3DSIM. Most notably, remote focusing was added to the microscope-aotools package. All changes were tracked and they are described in more detail in the respective branch repositories:

python-microscope: https://github.com/dstoychev/microscope/tree/deepsim

microscope-cockpit: https://github.com/dstoychev/cockpit/tree/deepsim

microscope-aotools: https://github.com/dstoychev/microscope-aotools/tree/deepsim-matthew-danny”

Minor comment #7Reported average values of the FWHM of imaged beads in 3D (p4) require also to report errors associated with those measurements.

We agree with this comment. To address this point, we have now removed the resolution numbers from the first paragraph of the Results section, but we revised Figure 2A extensively and now it contains comprehensive statistical information for the FWHM estimations.

Minor comment #8Page 13, second paragraph states that "The results from chapter 3.2…" I believe that was a copy/paste from a thesis but should be corrected for a peer-reviewed publication, as there is no chapter 3.2.

This was left over in error, from an older version of the supplementary material, which used numbered sectioning. To address this comment, we rephrased the sentence, which now correctly refers to Appendix 6:

“The findings from Appendix 6 demonstrated […]”